# Plasticity of *Escherichia coli* cell wall metabolism promotes fitness and antibiotic resistance across environmental conditions

Elizabeth A Mueller[1], Alexander JF Egan[2], Eefjan Breukink[3], Waldemar Vollmer[2], Petra Anne Levin[1]*

[1]Department of Biology, Washington University in St. Louis, St. Louis, United States; [2]Centre for Bacterial Cell Biology, Institute for Cell and Molecular Biosciences, Newcastle University, Newcastle upon Tyne, United Kingdom; [3]Membrane Biochemistry and Biophysics, Department of Chemistry, Faculty of Science, Utrecht University, Utrecht, Netherlands

**Abstract** Although the peptidoglycan cell wall is an essential structural and morphological feature of most bacterial cells, the extracytoplasmic enzymes involved in its synthesis are frequently dispensable under standard culture conditions. By modulating a single growth parameter—extracellular pH—we discovered a subset of these so-called 'redundant' enzymes in *Escherichia coli* are required for maximal fitness across pH environments. Among these pH specialists are the class A penicillin binding proteins PBP1a and PBP1b; defects in these enzymes attenuate growth in alkaline and acidic conditions, respectively. Genetic, biochemical, and cytological studies demonstrate that synthase activity is required for cell wall integrity across a wide pH range and influences pH-dependent changes in resistance to cell wall active antibiotics. Altogether, our findings reveal previously thought to be redundant enzymes are instead specialized for distinct environmental niches. This specialization may ensure robust growth and cell wall integrity in a wide range of conditions.
**Editorial note:** This article has been through an editorial process in which the authors decide how to respond to the issues raised during peer review. The Reviewing Editor's assessment is that all the issues have been addressed (see decision letter).
DOI: https://doi.org/10.7554/eLife.40754.001

*For correspondence:
plevin@wustl.edu

Competing interests: The authors declare that no competing interests exist.

## Introduction

The growth and survival of single-celled organisms relies on their ability to adapt to rapidly changing environmental conditions. A commensal, pathogen, and environmental contaminant, *Escherichia coli* occupies and grows in diverse environmental niches, including the gastrointestinal tract, bladder, freshwater, and soil. In the laboratory, the bacterium's flexibility in growth requirements is reflected in robust proliferation across a wide range of temperature, salt, osmotic, pH, oxygenation, and nutrient conditions (*Ingraham and Marr, 1996*).

The physiological adaptations that permit growth and survival across environmental conditions are not yet well understood, particularly for extracytoplasmic processes. Due to the discrepancy in permeability between the plasma and outer membrane (*Rosenbusch, 1990*), the periplasmic space of Gram-negative bacteria is sensitive to chemical and physical perturbations, including changes in salt, ionic strength, osmolality, and pH. Notably, upon mild environmental acidification, the periplasm assumes the pH of the extracellular media (*Slonczewski et al., 1981*; *Wilks and Slonczewski,*

*2007*). Although mechanisms that contribute to cytoplasmic pH homeostasis have been described in detail (*Castanie-Cornet et al., 1999*; *Castanié-Cornet et al., 2010*), comparatively little is known about the quality control mechanisms that preserve proper folding, stability, and activity of key proteins in the periplasm.

The peptidoglycan (PG) cell wall and its synthetic machinery are among the fundamental constituents of the periplasm that must be preserved across growth conditions. Essential for viability among most bacteria, PG is composed of glycan strands of repeating *N*-acetylglucosamine and *N*-acetylmuramic acid disaccharide units crosslinked at peptide stems (*Vollmer et al., 2008*). Beyond providing a force necessary to resist turgor pressure, the cell wall maintains cell shape, and components of the cell envelope serve as a major interface for cell-cell and cell-host interactions (*Typas et al., 2012*; *McDonald et al., 2005*). As an essential process, PG synthesis is also the principle target of several classes of antibacterial agents, including β-lactam (e.g. penicillin) and glycopeptide (e.g. vancomycin) antibiotics.

PG precursors are assembled in the cytosol and translocated across the inner membrane into the periplasm, where cell wall synthases construct the PG network through a series of glycosyltransferase (glycan polymerizing) and transpeptidase (peptide crosslinking) reactions. PG synthases include bifunctional class A penicillin binding proteins (PBPs), as well as monofunctional transpeptidases (class B PBPs) and monofunctional glycosyltransferases of the shape, elongation, division, and sporulation (SEDS) protein families (*Sauvage et al., 2008*; *Meeske et al., 2016*; *Cho et al., 2016*; *Taguchi et al., 2019*). LD-transpeptidases synthesize non-canonical LD-crosslinks between peptide stems. They are predominately active during PG remodeling during stationary phase growth in *E. coli* (*Pisabarro et al., 1985*; *Magnet et al., 2007*) and are required under severe envelope stress (*Morè et al., 2019*). In addition to synthases, a series of periplasmic cell wall hydrolases and autolysins—including DD-carboxypeptidases, DD and LD-endopeptidases, lytic transglycosylases, and amidases—are required to accommodate nascent strand insertion for expansion of the PG network, create substrate binding sites, and separate cells during the final stages of cytokinesis (*Typas et al., 2012*). These enzymes may also play a role activating synthases to ensure cell wall integrity (*Lai et al., 2017*).

The periplasmic steps of PG synthesis and remodeling exhibit high levels of enzymatic redundancy, the function of which remains unclear. While the cytoplasmic steps of PG precursor biogenesis in *E. coli* have nearly a 1:1 stochiometric ratio between reactions and enzymes (12 reactions: 14 enzymes), over 36 enzymes can carry out the nine reactions that take place in the periplasm (*Pazos et al., 2017*). Moreover, with the exception of the SEDS glycosyltransferase/bPBP pairs RodA/PBP2 and FtsW/PBP3 required for lateral expansion of the cell wall and cell division, respectively (*Meeske et al., 2016*; *Cho et al., 2016*), the remaining periplasmic cell wall enzymes appear to be nonessential. Inactivation of an individual enzyme—and in some cases, even multiple enzymes in the same class—often fails to confer discernable growth or morphological phenotypes under standard culture conditions (*Singh et al., 2012*; *Nelson and Young, 2000*; *Heidrich et al., 2002*; *Suzuki et al., 1978*; *Heidrich et al., 2001*). Although technological breakthroughs have aided in the identification of cell wall metabolic genes encoding proteins with similar functions (e.g. *Peters et al., 2016a*), elucidating the potential fitness benefit to redundancy has proven challenging.

One model to account for the apparent redundancy of periplasmic cell wall proteins is that enzymes within a given class may be specialists for distinct environmental niches, thereby allowing bacteria to cope with the diverse chemical and physical properties that might affect protein stability and function in this compartment (*Pazos et al., 2017*). In support of this hypothesis, several groups have identified cell wall enzymes that have increased activity in acidic media. Peters and colleagues demonstrated that *E. coli* carboxypeptidase PBP6b plays a key role in maintenance of cell morphology during growth at pH 5.0 (*Peters et al., 2016b*), while Castanheira *et al.* identified a PBP3 homolog in *Salmonella* Typhimurium that is preferentially involved in septation at low pH (*Castanheira et al., 2017*). Similarly, the lytic transglycosylase MltA exhibits maximal activity in acidic conditions in vitro (*van Straaten et al., 2005*), although whether this property is relevant in vivo remains unknown.

In light of these findings, we hypothesized that loss of an enzyme specialized for a particular environmental niche may produce a condition-specific growth defect through impaired cell wall integrity, allowing us to take a systems-level approach to identifying enzymes with differential roles in growth in vivo. In screening 32 mutants across six classes of nonessential periplasmic cell wall enzymes, we

determined that a subset of these enzymes is differentially required for fitness across pH environments. We find that disruptions in the activity of cell wall synthases PBP1a and PBP1b conferred fitness defects in opposing pH ranges that can be attributed in part to pH-dependent differences in enzymatic activity. We further demonstrate that synthase specialization has consequences for intrinsic resistance to β-lactam antibiotics in nonstandard growth conditions.

## Results

### Identification of pH specialist cell wall synthases and hydrolases

To determine the contribution of individual cell wall enzymes to pH-dependent growth, we cultured strains harboring deletions in genes encoding each of three class A PBPs, six LD-transpeptidases, five carboxypeptidases, four amidases, nine lytic glycosyltransferases, and six endopeptidases to mid-exponential phase ($OD_{600}$ ~0.2–0.6) in buffered LB media (pH 6.9) then sub-cultured them into fresh LB buffered to pH 4.8, 6.9, or 8.2 for growth rate analysis. These pH values were chosen as representative, physiologically relevant conditions *E. coli* cells encounter in the lower GI tract (pH 5–9) or urine (pH 4.5–8) (*Watson et al., 1972*; *Henderson and Palmer, 1912*). Preliminary hits were identified by a significant (>5%) decrease in early exponential phase ($OD_{600}$ 0.005–0.1) mass doublings per hour (DPH) in at least one pH condition compared to the parental strain. Representative growth curves and fits are presented in *Figure 1—figure supplement 1*. Mutants exhibiting significant growth defects at one or more pH values were re-tested across an expanded set of pH conditions for validation; those that displayed consistent growth defects across a discrete range of pH values were classified as pH-sensitive mutants (*Figure 1—figure supplement 2*; *Figure 2*; *Supplementary file 3*).

Collectively, five mutants met these stringent criteria and displayed significant pH-dependent reductions in DPH. We observed both acid-sensitive and alkaline-sensitive mutants across three enzymatic classes. Strikingly, loss of the bifunctional synthase PBP1b (*mrcB*) abolished growth at pH 4.8—over half a pH unit lower than the growth restrictive condition for the parental strain (*Figure 1A*; *Figure 1—figure supplement 3A*). Growth of the ΔmrcB mutant was also significantly attenuated (10–25% defect in DPH) at pH values between 5.1–5.9 but was indistinguishable from the parental strain in neutral and alkaline pH (*Figure 2A,C*). Pre-conditioning the mutant in acidic media (pH 5.5) did not abrogate the growth rate defect (*Figure 1—figure supplement 3A*), indicating that steady-state pH—rather than pH shock—underlies the defect in DPH. Mutants defective in production of lytic transglycosylase MltA and endopeptidase MepS (*spr*) also exhibited a specific, albeit less severe, defect in DPH in acidic media: their growth was attenuated by 5–15% compared to wild type cells at pH values at or below 6.2 (*Figure 1B,C*; *Figure 1—figure supplement 2*). Consistent with a role in acid tolerance, MltA was previously shown to have elevated enzymatic activity in acidic conditions in vitro (*van Straaten et al., 2005*), and cells defective for PBP1b or MepS production exhibit reduced colony growth on acidic agarose plates (*Nichols et al., 2011*).

We also identified two alkaline-sensitive mutants. Loss of the bifunctional synthase PBP1a (*mrcA*) and the lytic transglycosylase MltG (*yceG*) impaired, but did not abolish, growth specifically in neutral and alkaline media (*Figure 1A,C*; *Figure 1—figure supplement 2*; *Figure 2A,C*). Loss of MltG was associated with a greater magnitude and range of growth impairment (pH 6.2–8.4 compared to pH 6.5–8.2 for ΔmrcA). Both mutants' growth was restored to wild-type levels in acidic media (pH <6.0) and at pH 9.0 (*Figure 1—figure supplement 3B*).

Individual deletions in the genes encoding for the six LD-transpeptidases, five carboxypeptidases, and four amidases failed to confer any pH-dependent defects in DPH at any pH tested, consistent with either a limited role of these enzymes in exponential phase growth (*Pisabarro et al., 1985*; *Magnet et al., 2007*) or additional layers of redundancy (*Supplementary file 3*).

### Class A PBP activity ensures fitness across a wide pH range

Given their opposing impact on DPH under acidic and alkaline conditions, we elected to focus further efforts on understanding the contribution of the bifunctional class A PBPs PBP1a and PBP1b to growth across a range of pH conditions. An accumulating body of evidence suggests the class A PBPs play overlapping, and potentially redundant, roles in PG synthesis during growth in standard culture conditions (i.e. nutrient rich, neutral pH growth medium aerated at 37°C) (*Cho et al., 2016*;

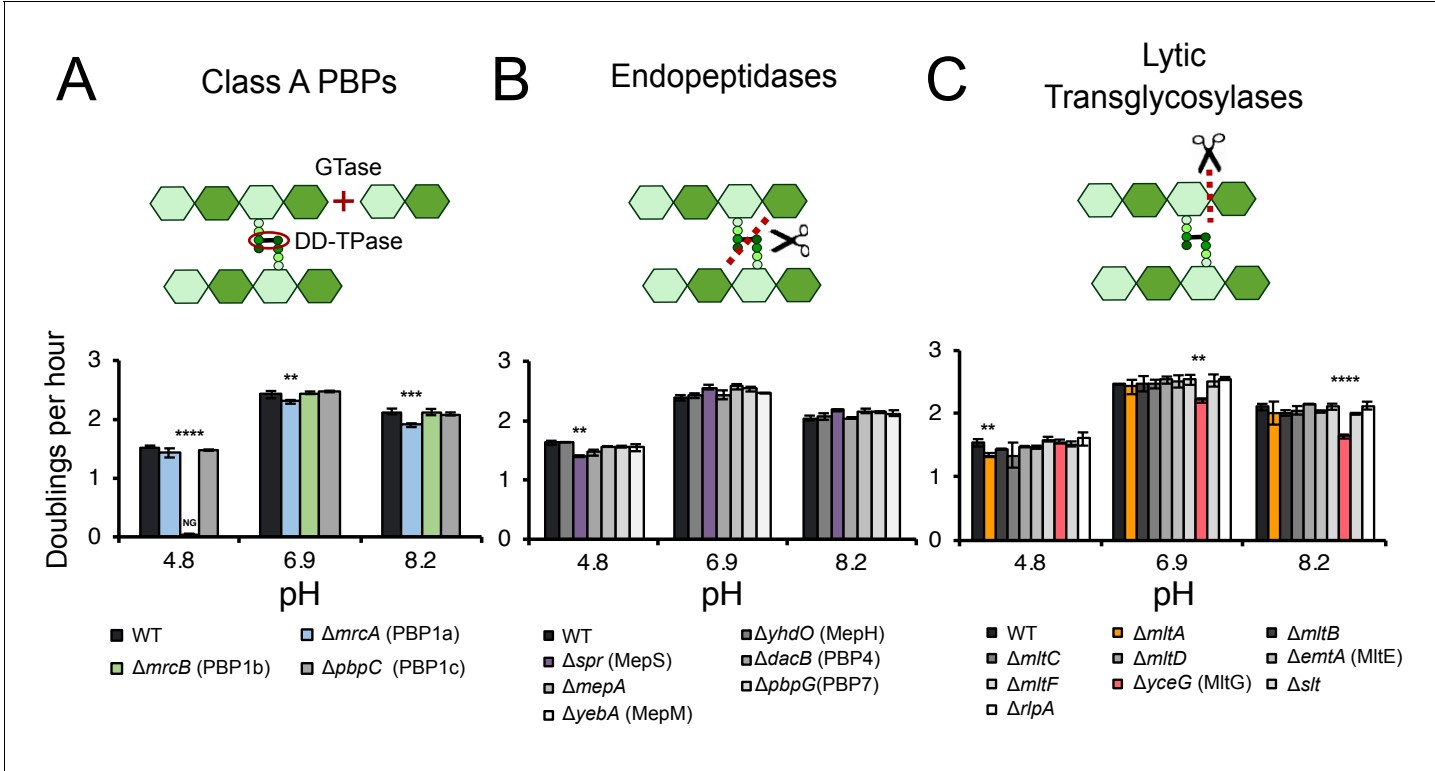

**Figure 1.** Identification of pH specialist cell wall enzymes. Mutants in genes encoding for non-essential class A PBPs (**A**), endopeptidases (**B**), and lytic transglycosylases (**C**) were screened for growth defects compared to the parental strain in LB media buffered to pH 4.8, 6.9, or 8.2. Bars depict mean mass doublings per hour ±SD of three independent biological replicates. NG denotes 'no growth' observed throughout the course of the experiment (20 hr). Cartoons depict enzymatic activity of the indicated enzyme class with GTase and DD-TPase referring to glycosyltransferase and DD-transpeptidase activity, respectively. Asterisks denote a significant >5% growth defect as determined by a one-way ANOVA corrected for multiple comparisons as follows: **, p<0.01; ***, p<0.001; ****p<0.0001. Mean ± SD values for mutants in this figure and all additional mutants tested can be viewed in *Supplementary file 3*. Representative growth curves, fits, and source data from the class A PBP mutants can be viewed in *Figure 1—figure supplement 1* and *Figure 1—source datas 1–3*.

DOI: https://doi.org/10.7554/eLife.40754.002

The following source data and figure supplements are available for figure 1:

**Source data 1.** Representative source data for class A PBP mutants at pH 4.8.
DOI: https://doi.org/10.7554/eLife.40754.006

**Source data 2.** Representative source data for class A PBP mutants at pH 6.9.
DOI: https://doi.org/10.7554/eLife.40754.007

**Source data 3.** Representative source data for class A PBP mutants at pH 8.2.
DOI: https://doi.org/10.7554/eLife.40754.008

**Figure supplement 1.** Growth rate determination pipeline.
DOI: https://doi.org/10.7554/eLife.40754.003

**Figure supplement 2.** Growth rate determination of hits across an expanded set of pH conditions.
DOI: https://doi.org/10.7554/eLife.40754.004

**Figure supplement 3.** Growth of class A PBP mutants in extreme pH conditions.
DOI: https://doi.org/10.7554/eLife.40754.005

*Yousif et al., 1985*). Indeed, *E. coli* requires at least one of these enzymes for viability during growth in standard culture conditions (*Suzuki et al., 1978*).

Based on their disparate pH-dependent growth defects, we hypothesized that PBP1a and PBP1b are specialized synthases whose activity is essential for maximal growth in distinct pH environments. Consistent with this model, cells defective in PBP1a (Δ*mrcA*) and PBP1b (Δ*mrcB*) displayed defects in DPH at discrete, non-overlapping pH ranges (*Figure 2A–C*). Loss of PBP1c (*pbpC*), a third class A PBP with an unclear role in cell wall metabolism (*Schiffer and Höltje, 1999*), did not result in a defect in DPH at any pH tested alone or combination with cells defective for PBP1a or PBP1b

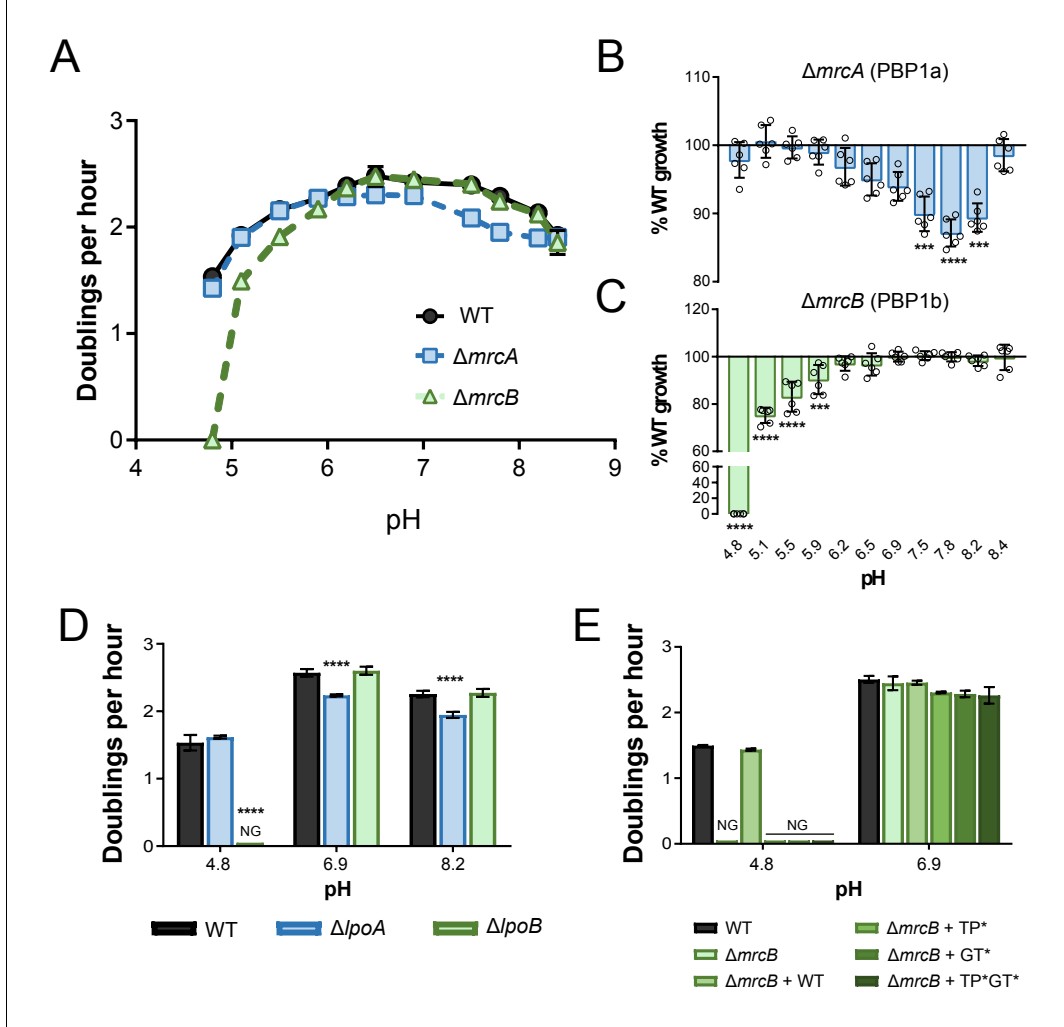

**Figure 2.** pH-dependent growth requires class A PBP activity. (**A–C**) Mean mass doublings per hour and transformed percent parental growth for ΔmrcA (PBP1a; EAM543) and ΔmrcB (PBP1b; EAM546) deletions compared to parental strain (MG1655) in LB media buffered from pH 4.8–8.4 Significance was determined by an unpaired t-test corrected for multiple comparisons using the Holm-Sidak method. Error bars represent SD from six independent biological replicates. (**D**) Growth rate analysis of cells defective for LpoA (EAM657) and LpoB (EAM659) cultured in buffered LB at pH 4.8, 6.9, or 8.0. Bars represent mean mass doublings per hour ± SD from three independent biological replicates. Asterisks denote significance as determined by a one-way ANOVA corrected for multiple comparisons as follows: ****, p<0.0001. Growth of these mutants across an expanded set of pH conditions can be viewed in *Figure 2—figure supplement 2*. (**D**) Complementation analysis of PBP1b variants synthesized from a plasmid (pUM1Bα, pUM1Bα*, pUM1BTG*α, or pUM1BTG*α*) and induced with 5 μM IPTG in the ΔmrcB (EAM696) background in buffered LB at pH 4.8 and 6.9. Bars represent mean mass doublings per hour ± SD from three independent biological replicates. NG denotes 'no growth' observed throughout the course of the experiment (20 hr).
DOI: https://doi.org/10.7554/eLife.40754.009
The following figure supplements are available for figure 2:

**Figure supplement 1.** Growth rate analysis of strains defective for PBP1c.
DOI: https://doi.org/10.7554/eLife.40754.010
**Figure supplement 2.** Growth rate determination of *lpo* mutants across an expanded set of pH conditions.
DOI: https://doi.org/10.7554/eLife.40754.011

(*Figure 2—figure supplement 1*), indicating this enzyme does not play a role in pH-dependent growth under the conditions tested here.

We next sought to test whether PBP transpeptidase and/or glycosyltransferase activity were required for fitness across pH conditions, as opposed to an indirect, structural role for these enzymes in the formation of cell wall synthesis complexes (*Müller et al., 2007*; *Bertsche et al., 2006*). To test this, we took advantage of two sets of mutants: 1) deletions in *lpoA* and *lpoB*—genes encoding outer membrane lipoprotein cofactors required for activity, but not expression or stability, of PBP1a and PBP1b, respectively (*Typas et al., 2010*; *Paradis-Bleau et al., 2010*; *Egan et al., 2014*; *Lupoli et al., 2014*), and 2) point mutations that inactivate PBP1b transpeptidase and/or glycosyltransferase activity but do not impact stability (*Meisel et al., 2003*).

Implicating PBP activity in growth across pH environments, loss of the cofactors LpoA and LpoB mimicked the pH-dependent growth defects of loss of the enzymes themselves. Analogous to cells defective for PBP1b, deletion of *lpoB* prevented growth at pH 4.8. Likewise, loss of PBP1a's cofactor LpoA led to a significant defect in DPH between pH values 5.9–8.2 (*Figure 2D*; *Figure 2—figure supplement 2*). Interestingly, loss of either Lpo protein did not perfectly recapitulate loss of its cognate class A PBP (*Figure 2—figure supplement 2*; *Figure 2A–C*), suggesting the presence of additional relevant regulators in vivo. Complementation analysis of PBP1b variants at acidic pH further bolstered the conclusion that activity is required for pH-dependent growth. As expected, production of wild-type PBP1b in trans restored growth of the ΔmrcB mutant at pH 4.8; however, production of PBP1b variants rendering the transpeptidase (S510A, TP*), glycosyltransferase (E233Q, GT*), or both enzymatic activities inactive (TP*GT*) failed to complement growth (*Figure 2E*). It should be noted that the mutation in the glycosyltransferase active site (E233Q) previously has been shown to attenuate transpeptidase activity by 90%, consistent with observations that PBP transpeptidase activity cannot be assayed in vitro in the absence of functional glycosyltransferase activity (*Egan et al., 2014*; *Bertsche et al., 2005*; *Terrak et al., 1999*; *Born et al., 2006*). Thus, although our data demonstrate that transpeptidase activity is critical for pH-dependent growth, we cannot discern whether glycosyltransferase activity alone is required.

## Class A PBP activity promotes cell wall integrity across pH environments

Although these findings establish PBP1a and PBP1b activity as essential for optimal fitness across a wide pH range, it remained unclear whether these mutants' pH-dependent defects in DPH in bulk culture were due to reduction in growth across the population (i.e. decreased rate of mass accumulation and cell expansion) or lysis of a fraction of cells in the population. To differentiate between these two mechanisms, we inoculated early exponential phase ($OD_{600}$ ~0.05–0.1) wild type or mutant cells cultured at pH 6.9 on to agarose pads buffered to pH 4.5 or pH 8.0 and examined cells for incorporation of the dye propidium iodide (PI), which permeates cells with compromised membranes, by microscopy.

Consistent with a lytic origin, extensive PI incorporation was observed at one hour post-shift for PBP1b and PBP1a defective cells that underwent acid (pH 6.9 to pH 4.5) or alkaline (pH 6.9 to pH 8.0) shock, respectively (*Figure 3C,D*). To confirm *bona fide* lysis, we transformed a plasmid expressing *gfp* in to the mutants and measured PI incorporation and loss of cytoplasmic GFP concurrently. Without exception, PI+ cells lacked cytoplasmic GFP signal (*Figure 3—figure supplement 1*).

Time lapse imaging of cells following pH shift shed light on lysis kinetics: upon pH downshift, ΔmrcB cells began to incorporate PI by 30 min (~15% cells labeled), with ~95–100% of cells labeled by two hours post-shift. Negligible cell death was observed for the parental strain or for cells defective for PBP1a during equivalent acid shock (*Figure 3C*). Conversely, up to 15% of ΔmrcA cells underwent lysis an hour following alkaline shift to pH 8.0 (*Figure 3D*). Although we did observe reduced rates of lysis for the ΔmrcA mutant at later time points, this recovery was not recapitulated in liquid culture and thus was not investigated further (*Figure 3—figure supplement 1D*). ΔmrcB and wild type cells exhibited minimal (<5%) or negligible cell death, respectively, in response to alkaline shift on agarose pads. Prior to cell lysis, single cell growth rate of both mutants and the parental strain were similar at each pH condition (*Figure 3A,B*), indicating lysis is the sole determinant of decreased DPH observed in bulk culture.

In addition to displaying differential lysis kinetics under their respective non-permissive pH conditions, PBP1a and PBP1b mutants also differed in apparent lysis mechanism. Time lapse imaging during acid shock revealed a high fraction of ΔmrcB cells lysed during division, often from a bulge emanating at the septum (*Figure 4A*, white arrows; *Figure 4—video 1*). Scanning electron

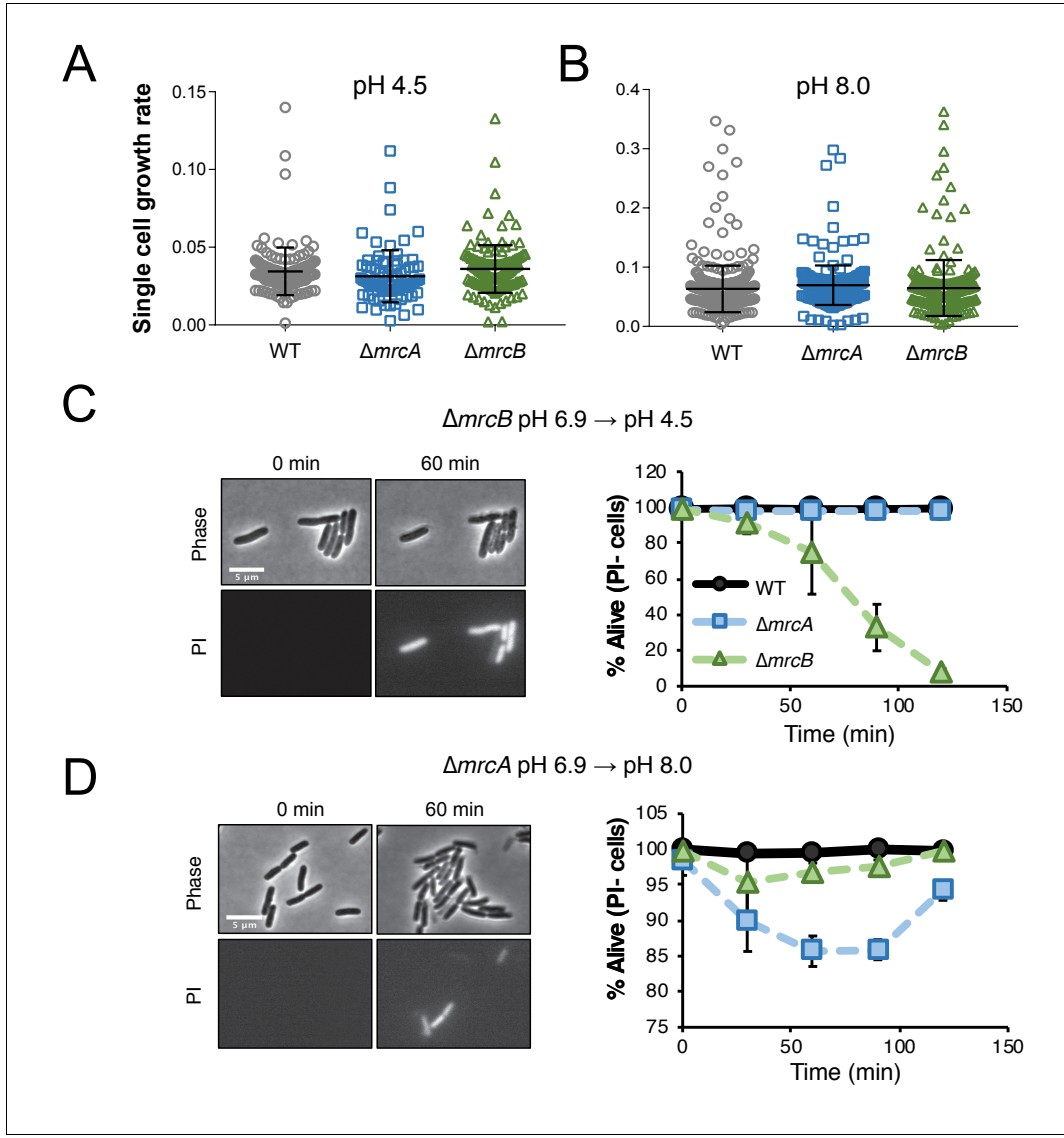

**Figure 3.** Cells defective for class A PBPs lyse upon exposure to non-permissive pH conditions. (A–B) Single cell elongation rates for wild-type (MG1655), ΔmrcA (EAM543), and ΔmrcB (EAM546) cells during growth on agarose pads at pH 4.5 (A; n = 134, 81, and 155 cells) and pH 8.0 (B; n = 386, 257, and 246 cells). Rates were determined in the MATLAB-based program SuperSegger as described in the methods. Error bars represent SD. (C–D) Micrographs depicting representative images of propidium iodide incorporation in ΔmrcA (D, left) and ΔmrcB (C, left) mutants at t = 0 or 60 min post indicated pH shift. Scale bar represents 5 µm. Cell viability curves for wild-type, ΔmrcA (PBP1a), and ΔmrcB (PBP1b) strains after acidic (D, right) or alkaline pH (C, right) shift as indicated. Cell death was determined by uniform cytoplasmic staining with propidium iodide. Markers indicate mean percent viability ± SD of three biological replicates. Greater than 100 cells were analyzed for each strain at each time point per replicate.

DOI: https://doi.org/10.7554/eLife.40754.012

The following figure supplement is available for figure 3:

**Figure supplement 1.** Cytoplasmic GFP loss correlates with propidium iodide staining.

DOI: https://doi.org/10.7554/eLife.40754.013

microscopy confirmed the bulges were coincident with the septum in this mutant (*Figure 4C*). To quantitate the lytic phenotype of the mutants, we categorized the lysis mechanism into three groups: septal bulge, non-septal bulge (including polar and peripheral bulging), and lysis not associated with visible bulging. Septal bulging was determined based on association of the bulge origin

with the visible constriction site by phase contrast microscopy. Indeed, this analysis confirmed our observation: 55% of Δ*mrcB* mutants lysed at the septum following pH downshift with the remaining fraction associated with a non-septal bulge (17%) and no bulge (28%) (*Figure 4D*). In contrast, lysis of the Δ*mrcA* mutant during growth in alkaline pH was not associated with division, and instead, lysis of ~60% of the cells was coincident with a non-septal bulge, typically emanating from the periphery (*Figure 4B,D*; *Figure 4—video 2*). These differences may reflect distinct localization preferences of the enzymes (*Bertsche et al., 2006*; *Banzhaf et al., 2012*) or disparate weak regions in the PG across pH conditions.

## PBP1a localization and activity are impaired at low pH

Although our data support a model in which class A PBP activity is differentially required for cell wall integrity across pH environments, the mechanistic basis for pH specialization remained unclear. To interrogate this, we compared the production, localization, and biochemical activity of PBP1a and PBP1b as a function of pH.

We predicted differential PBP production across pH conditions may contribute to the enzymes' specialization, as has previously been shown for acid-specialist carboxypeptidase PBP6b (*Peters et al., 2016b*). Consistent with previous proteomic mass spectrometry data (*Schmidt et al., 2016*), bulk protein levels of both class A PBPs were modestly reduced in acidic media (*Figure 5C*; *Figure 5—figure supplement 1*). However, pH-dependent differences in production did not suggest a correlation between either class A PBP's levels and its contribution to fitness in a particular pH environment.

We next turned to examining PBP localization across pH conditions, using functional GFP fusions to PBP1a and PBP1b produced from the *attHK* locus and under IPTG inducible control (*Paradis-Bleau et al., 2010*). Similar to previous reports (*Bertsche et al., 2006*; *Paradis-Bleau et al., 2014*), the fusion proteins exhibited discrete localization profiles at neutral pH: GFP-PBP1a localized predominantly to the cell periphery, while GFP-PBP1b was present at both the periphery and the septum. Although GFP-PBP1b localization did not noticeably differ across pH conditions, GFP-PBP1a peripheral signal was reduced at pH 5.0, and the fusion adopted an irregular clustered distribution throughout the cell body (*Figure 5A,B*; *Figure 5—figure supplement 2*). The physiological significance of this phenotype remains unclear and requires additional investigation.

To examine the effect of pH on PBP synthase activity, we first utilized an end-point assay that concurrently measures glycosyltransferase activity and transpeptidase domain activity, which are coupled in the bifunctional class A PBPs (*Bertsche et al., 2005*; *Born et al., 2006*; *Egan et al., 2018*). Although purified PBP1a and PBP1b exhibit biochemical activity alone in vitro, we chose to test the influence of pH on the class A PBPs in the context of their key cellular activators, including LpoA for PBP1a and LpoB and FtsN for PBP1b. The outer membrane lipoprotein activators LpoA and LpoB are required for the function of their cognate class A PBP in vivo (*Typas et al., 2012*; *Paradis-Bleau et al., 2010*), and the essential division protein FtsN can act synergistically with LpoB to enhance PBP1b glycosyltransferase activity up to 16-fold in vitro (*Egan et al., 2015*).

Briefly, purified enzymes and their cognate activators were solubilized into detergent micelles and incubated with [$^{14}$C]lipid II precursor in buffer at pH 4.8, 6.9, and 8.2. After one hour, the resulting PG was digested into muropeptides and resolved by high performance liquid chromatography. Glycosyltransferase activity is reflected qualitatively in the proportion of [$^{14}$C]lipid II utilized, resulting in a decrease in peak 1. Transpeptidase activity—including both crosslinking activity and carboxypeptidase activity—is quantified by the fraction of muropeptides with modified peptides (peaks 5–9) (*Egan et al., 2015*). Strikingly, at pH 4.8 PBP1a + LpoA exhibited little glycosyltransferase or transpeptidase domain activity; in this condition the majority of [$^{14}$C]lipid II precursor was not polymerized (*Figure 5D*, peak 1). In contrast, PBP1b + LpoB + FtsN maintained similar end-point activity across all tested pH conditions (*Figure 5D,E*). Low PBP1a glycosyltransferase activity at pH 4.8 was confirmed in a continuous fluorescence assay, which measures the polymerization rate of a Dansyl-labeled Lipid II substrate. Consistent with previous work on PBP1b (*Egan et al., 2014*), in the absence of their cognate activators, both PBP1a and PBP1b exhibited reduced rates of glycosyltransferase activity in acidic conditions. However, co-incubation of PBP1b with LpoB and division protein FtsN significantly increased its polymerization rate under all pH conditions, while PBP1a exhibited poor activity even in the presence of LpoA (*Figure 5—figure supplement 3*).

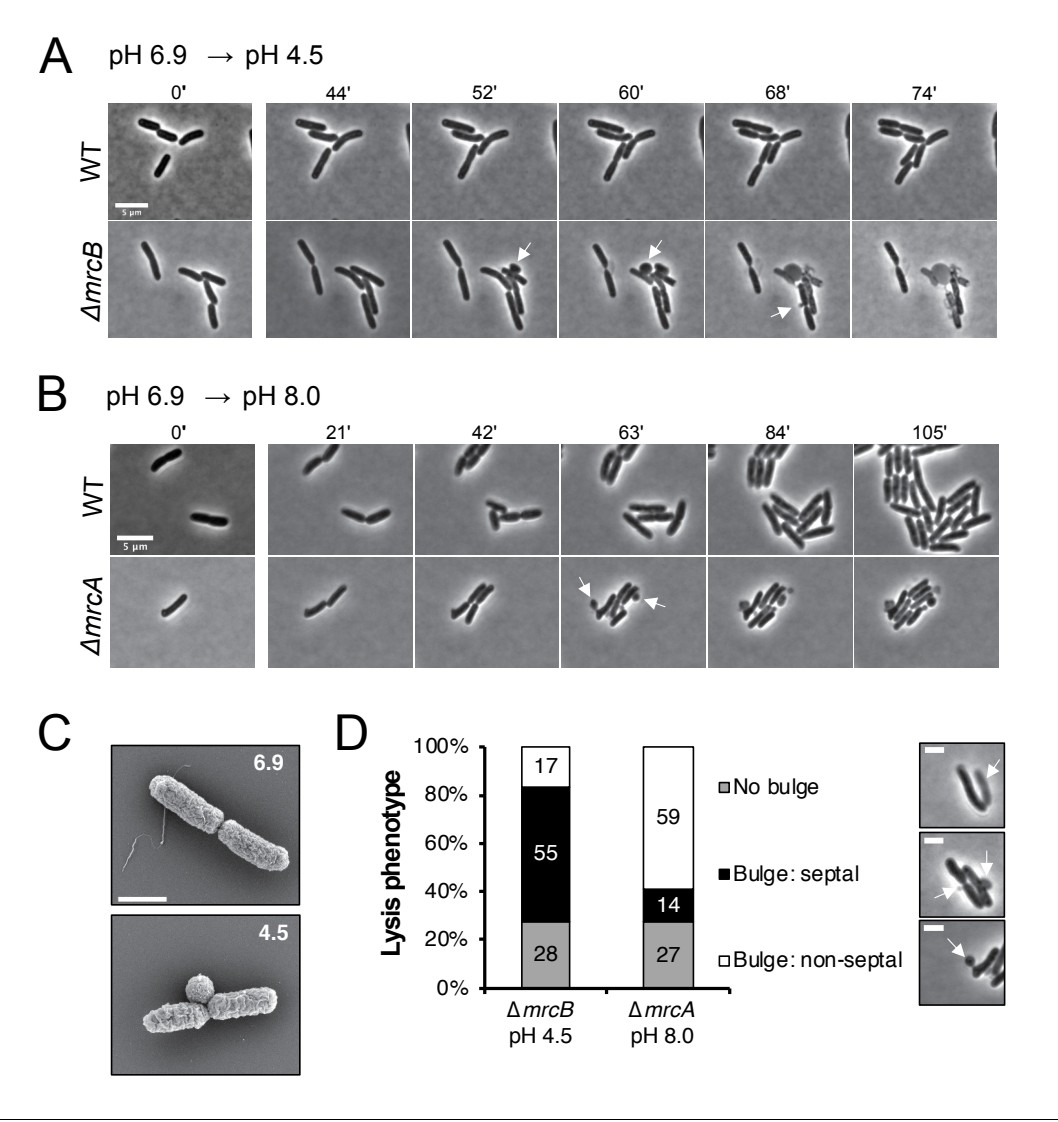

**Figure 4.** Distinct lytic phenotypes for cells defective for PBP1a and PBP1b upon pH shift. (A–B) Representative phase contrast frames of time lapse imaging of ΔmrcB (PBP1b; EAM546) and ΔmrcA (PBP1a; EAM543) mutants upon acidic (A) or alkaline (B) pH shift, respectively, as compared to the parental strain (MG1655). White arrows indicate membrane bulges. Scale bar denotes 5 μm. Full videos can be viewed in *Figure 4—videos 1* and *2*. (C) Representative scanning electron microscopy micrographs for ΔmrcB (PBP1b; EAM546) mutant shifted to either pH 6.9 or pH 4.5 for two hours prior to fixation. Scale bar represents 1 μm. (D) Quantification of lysis phenotype between mutants. Lytic terminal phenotype was categorized into three groups: lysis via septal bulge, non-septal bulge, or no bulge. Determination of lytic phenotype was based on the frames preceding propidium iodide incorporation (time step = 3 min). Micrographs (top to bottom) depict representative images of no bulge, septal bulge, and non-septal bulge, respectively with arrows (scale bar = 2 μm). At least 50 cells across at least two independent biological replicates were assessed (ΔmrcA, n = 128; ΔmrcB, n = 278 cells). Bars are subdivided based on percent lytic phenotype in each mutant.

DOI: https://doi.org/10.7554/eLife.40754.014

The following videos are available for figure 4:
**Figure 4—video 1.** ΔmrcB cells undergo septal lysis upon exposure to acidic media.

DOI: https://doi.org/10.7554/eLife.40754.015

**Figure 4—video 2.** Subpopulation of ΔmrcA cells lyse upon exposure to alkaline media.

DOI: https://doi.org/10.7554/eLife.40754.016

To test whether pH-dependent changes in PBP activity reflect a change in affinity of the enzymes for their cognate lipoprotein activators, we performed surface plasmon resonance experiments in which PBP1a and PBP1b were immobilized to chips and exposed to LpoA or LpoB at various concentrations. PBP1a-LpoA and PBP1b-LpoB bound at $K_D$ values of 520 ± 49 nM and 213 ± 22 nM at pH 6.9, respectively. Both $K_D$ values were ~2 fold higher at pH 8.2 (*Figure 5—figure supplement 4*), but affinity values could not be determined at pH 4.8 due to significant non-specific binding of both LpoA and LpoB to the chip. Altogether, our data support a model in which PBP1a activity is reduced in acidic conditions, rendering the cell reliant on PBP1b for cell wall integrity and viability.

## Low pH promotes intrinsic resistance to PBP2 and PBP3-specific β-lactams

Combined with the recent findings of other groups (*Peters et al., 2016b*; *Castanheira et al., 2017*; *Montón Silva et al., 2018*), our results suggest that the active cell wall synthesis machinery varies across pH environments. We hypothesized that one potential consequence of environmental plasticity in the cell wall synthesis machinery may be changes in intrinsic resistance to cell wall active antibiotics. If true, condition-dependent intrinsic resistance may have important implications for treatment of *E. coli* infections in host niches with variable pH (*Watson et al., 1972*; *Henderson and Palmer, 1912*). To test this model, we measured the minimum inhibitory concentration (MIC) of a panel of compounds during growth of *E. coli* strain MG1655 in a range of physiologically relevant pH conditions (pH 4.5–8.0). We focused on the β-lactam class of antibiotics, which often target specific PBPs at a drug's MIC (*Kocaoglu and Carlson, 2015*).

In support of our hypothesis, we observed a 4 to 32-fold increase in MIC to a subset of the tested β-lactams at pH values < 6.0 (*Figure 6A*; *Supplementary file 4*). In particular, cells displayed an increase in intrinsic resistance to compounds that specifically target PBP2 and PBP3, class B PBPs that are essential for cell elongation and division, respectively (*Kocaoglu and Carlson, 2015*). Consistent with acidic pH conferring a protective effect on the elongation and division machinery and previous reports (*Goodell et al., 1976*), cells cultured in low pH media retained near-normal morphology in the presence of concentrations of the compounds that led to either filamentation (cephalexin, CEX) or cell rounding (mecillinam, MEC) at pH 7.0 (*Figure 6B,C*; *Figure 6—figure supplement 2*). The pH-dependent change in intrinsic resistance was not limited to our laboratory strain: uropathogenic *E. coli* isolate UTI89 (*Chen et al., 2006*) exhibited a comparable change in MIC to both cephalexin (CEX) and mecillinam (MEC) at low pH during growth in both broth culture and in urine (*Figure 6D*; *Supplementary file 5*). In contrast, susceptibility to non-specific β-lactams (ampicillin, AMP; amoxicillin, AMX), a class A PBP-targeting compound (cefsulodin, CFS) (*Kocaoglu and Carlson, 2015*), or a protein synthesis inhibitor (chloramphenicol, CH) was not strongly pH-dependent. To rule out alternative causes of antibiotic resistance, we confirmed that differences in drug susceptibility could not be attributed to pH-dependent changes in antibiotic stability, proton motive force, β-lactamase production, or outer membrane permeability (*Figure 6—figure supplements 1* and *3*). These findings suggest that pH-mediated plasticity in the cell wall synthesis machinery influences intrinsic β-lactam sensitivity.

## PBP1b is required for low pH-dependent β-lactam resistance

We next sought to identify which, if any, cell wall enzymes were required for resistance to PBP2 and PBP3-targeting compounds in acidic media. We narrowed our focus to three classes of enzymes: 1) non-essential transpeptidases, including the class A PBPs and the LD-transpeptidases, which have been implicated in β-lactam resistance (*Lai et al., 2017*; *Hugonnet et al., 2016*; *Peters et al., 2018*); 2) pH specialist autolysins identified in this work; and 3) PBP6b, a carboxypeptidase required for proper morphology—but not growth—in acidic media (*Peters et al., 2016b*). Mutants defective for production of each enzyme were tested for loss of resistance to CEX at pH 5.5 compared to pH 7.0.

Genetic analysis suggests that PBP1b activity is specifically required for pH-dependent resistance to β-lactam antibiotics. Strains defective for production of PBP1a, PBP1c, LdtD, LdtE, MepS, MltA, MltG, and PBP6b exhibited a similar increase in resistance to CEX at pH 5.5 as the parental strain (*Figure 6—figure supplement 4A*). In contrast, loss of PBP1b abolished resistance at pH 5.5 and in fact, slightly increased susceptibility to CEX (*Figure 6E*). This phenotype was not specific to CEX:

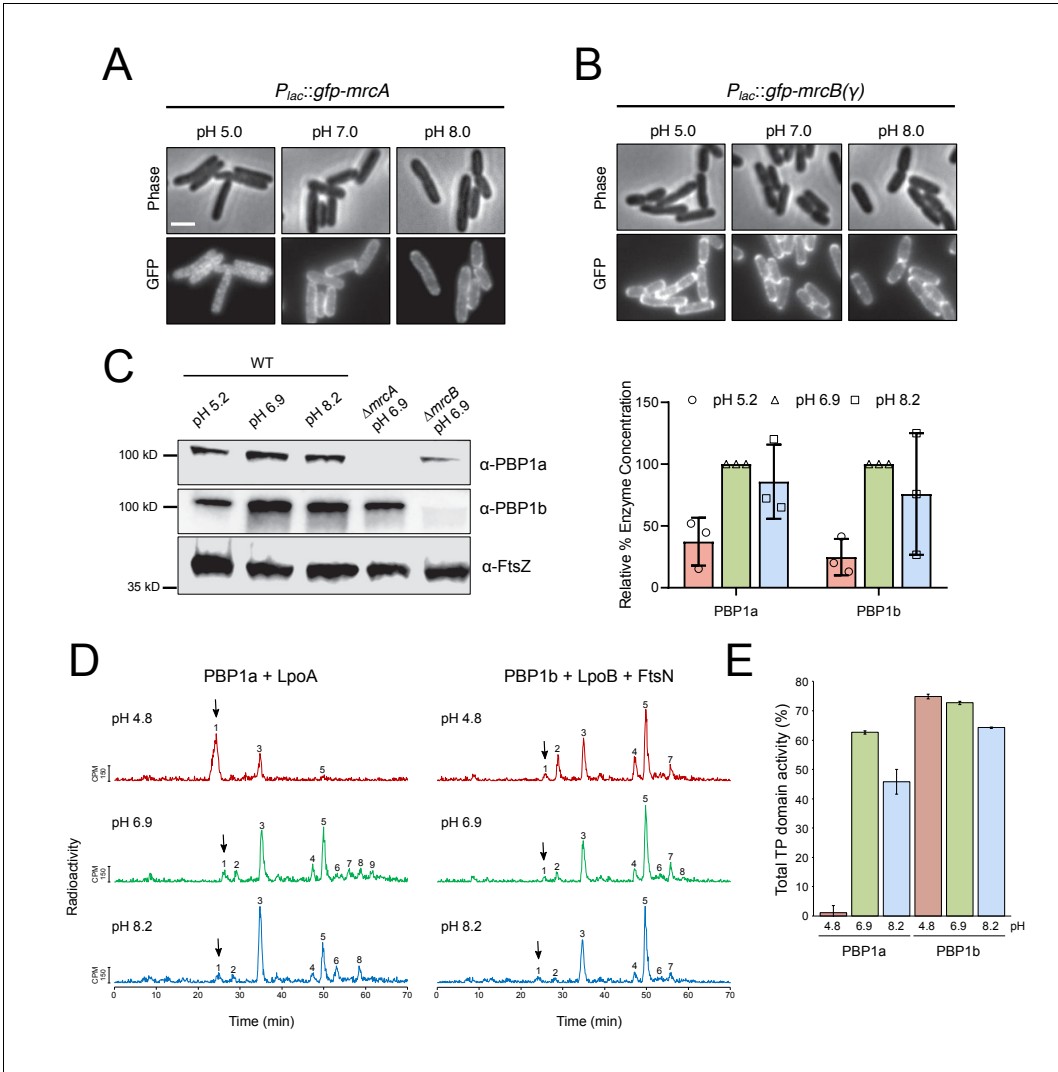

**Figure 5.** pH-dependence of aPBP localization, production and activity. (**A, B**) Representative micrographs illustrating aPBP localization in strains expressing $P_{lac}::gfp-mrcA$ (EAM707) and $P_{lac}::gfp-mrcB$ (EAM718) grown in AB minimal media supplemented with 0.2% maltose and 250 µM IPTG at pH 5.0, 7.0 and 8.0. Scale bar indicates 2 µm. (**C**) Western blot depicting representative biological replicates of PBP1a, PBP1b, and FtsZ levels in wild-type cells (MG1655) cultured at pH 4.8, 6.9, and 8.2. Percent aPBP level (using FtsZ levels as an internal loading control and normalized to pH 6.9 values) across pH conditions is shown to the right. Ponceau staining for total protein levels can be viewed in *Figure 5—figure supplement 1*. (**D**) Representative HPLC chromatograms of muropeptide analysis. Peak 1 (black arrows), Penta-P (stems from remaining lipid II/glycan chain ends); peak 2; Tetra (GT and CPase product); peak 3, Penta (GTase product); peak 4, TetraTetra (GTase, TPase and CPase product); peak 5, TetraPenta (GTase and TPase product), peak 6, TetraTetraTetra (GTase, TPase and CPase product); peak 7, TetraTetraPenta (GTase and TPase product); peak 8, TetraTetraTetraTetra (GTase, TPase and CPase product). (**E**) Quantification of TPase domain activity (sum of TPase and CPase products) of PBP1A + LpoA and PBP1B + LpoB and FtsN at pH 4.8, 6.9 and 8.2. Data is the mean ± range of two replicates. Corresponding representative HPLC chromatograms are shown in D.
DOI: https://doi.org/10.7554/eLife.40754.017

The following figure supplements are available for figure 5:

**Figure supplement 1.** Total protein quantification of class A PBP Western Blot.
DOI: https://doi.org/10.7554/eLife.40754.018

**Figure supplement 2.** PBP1a peripheral localization across pH conditions.
DOI: https://doi.org/10.7554/eLife.40754.019

**Figure supplement 3.** Influence of pH on class A PBP polymerization rate in continuous fluorescence glycosyltransferase assay.

*Figure 5 continued on next page*

*Figure 5 continued*

DOI: https://doi.org/10.7554/eLife.40754.020

**Figure supplement 4.** Influence of pH on PBP-lpo binding affinity.

DOI: https://doi.org/10.7554/eLife.40754.021

resistance to other PBP2 and PBP3 targeting compounds was also eliminated or significantly reduced in cells defective for PBP1b (*Figure 6—figure supplement 4C*). PBP1b enzymatic activity was required for resistance. Loss of the enzyme's cognate outer membrane lipoprotein LpoB or inactivation of its catalytic activity eliminated CEX resistance at low pH (*Figure 6E*; *Figure 6—figure supplement 3B*; *Supplementary file 6*). Importantly, a mutant with a comparable growth rate defect at pH 5.5 (Δ*mrcB* 1.91 ± 0.02 DPH; Δ*tolA* 1.40 ± 0.04 DPH) still displayed the same fold change in resistance to CEX at pH 5.5 as the parental strain. Likewise, a mutant in PBP5 (Δ*dacA*) with increased sensitivity to CEX even at neutral pH, similar to Δ*mrcB* (*Schmidt et al., 1981*; *García del Portillo and de Pedro, 1990*), also retained the resistance phenotype in acidic growth conditions (*Sarkar et al., 2010*) (*Figure 6—figure supplement 3D*). In sum, our findings point to a specific role for PBP1b in intrinsic β-lactam resistance in acidic media.

## Discussion

### Class A PBPs protect cell wall integrity across environmental conditions

By varying a single growth parameter—extracellular pH—we uncovered specialized roles for a subset of nonessential cell wall synthases and autolysins in *E. coli* that previously had been classified as redundant for growth. Of the pH specialist enzymes identified, we focused on the bifunctional synthases PBP1a and PBP1b, which we find are required for cell wall integrity in distinct pH environments. Failure to produce PBP1b in acidic media (pH <5.9) or PBP1a in more alkaline conditions (pH 6.5–8.2) reduced fitness and led to cell lysis (*Figure 2* and *Figure 3*). This lytic death is characteristic of class A PBP-deficient cells (*Suzuki et al., 1978*) and is consistent with their essential role in PG synthesis, perhaps by filling gaps in the PG foundation (*Cho et al., 2016*). Importantly, a recent study failed to observe major differences in global PG composition in *E. coli* cells grown in pH 7.5 and pH 5.0 (*Peters et al., 2016b*). Hence, the differential requirement for the class A PBPs in growth across pH conditions is unlikely to be a consequence of altered cell wall structure or distinct enzymatic activity of the PBPs.

Instead, our data suggest that pH-dependent differences in class A PBP activity underlie their contribution to maximal fitness in distinct pH environments. Even in the presence of its activator LpoA, PBP1a exhibits little glycosyltransferase and transpeptidase activity in vitro in acidic conditions, rendering the cell reliant on PBP1b to provide the essential class A PBP activity (*Figure 5D,E*; *Figure 5—figure supplement 3*). At present, the precise mechanism for reduced activity of PBP1a in acidic media remains unclear; possible causes include pH-dependent changes in structure, substrate binding, or catalysis, as well as reduced affinity to LpoA in acidic conditions. We attempted to test the latter model using surface plasmon resonance (SPR), but technical limitations prevented us from drawing firm conclusions (*Figure 5—figure supplement 4*). Although further work is needed, the depletion of PBP1a from the cell periphery in acidic media may also reflect an inactive state of the protein in vivo (*Figure 5A*).

Unlike PBP1a, PBP1b activity is largely invariant in vitro across the pH conditions tested when assayed in the presence of its key regulators (*Figure 5D,E*; *Figure 5—figure supplement 3*). Although this finding is consistent with our observation that PBP1b can partially compensate for loss of PBP1a in alkaline media (*Figure 2*), it remains unclear why PBP1a is required for maximal fitness in alkaline media. Additional factors in vivo, such as novel class A PBP regulators, may be responsible for the cell's preference for PBP1a in alkaline conditions that are not recapitulated in our in vitro assays. In support of this model, PBP1b exhibits reduced binding to radiolabeled penicillin in membrane extracts incubated in alkaline buffer relative to neutral or acidic conditions (*Amaral et al., 1986*). Alternatively, differences in the enzymes' localization and/or interaction partners may also play a role in their pH specialization. For example, PBP1a and PBP1b preferentially associate with specialized cell wall synthesis complexes essential for cell elongation and cell division, respectively

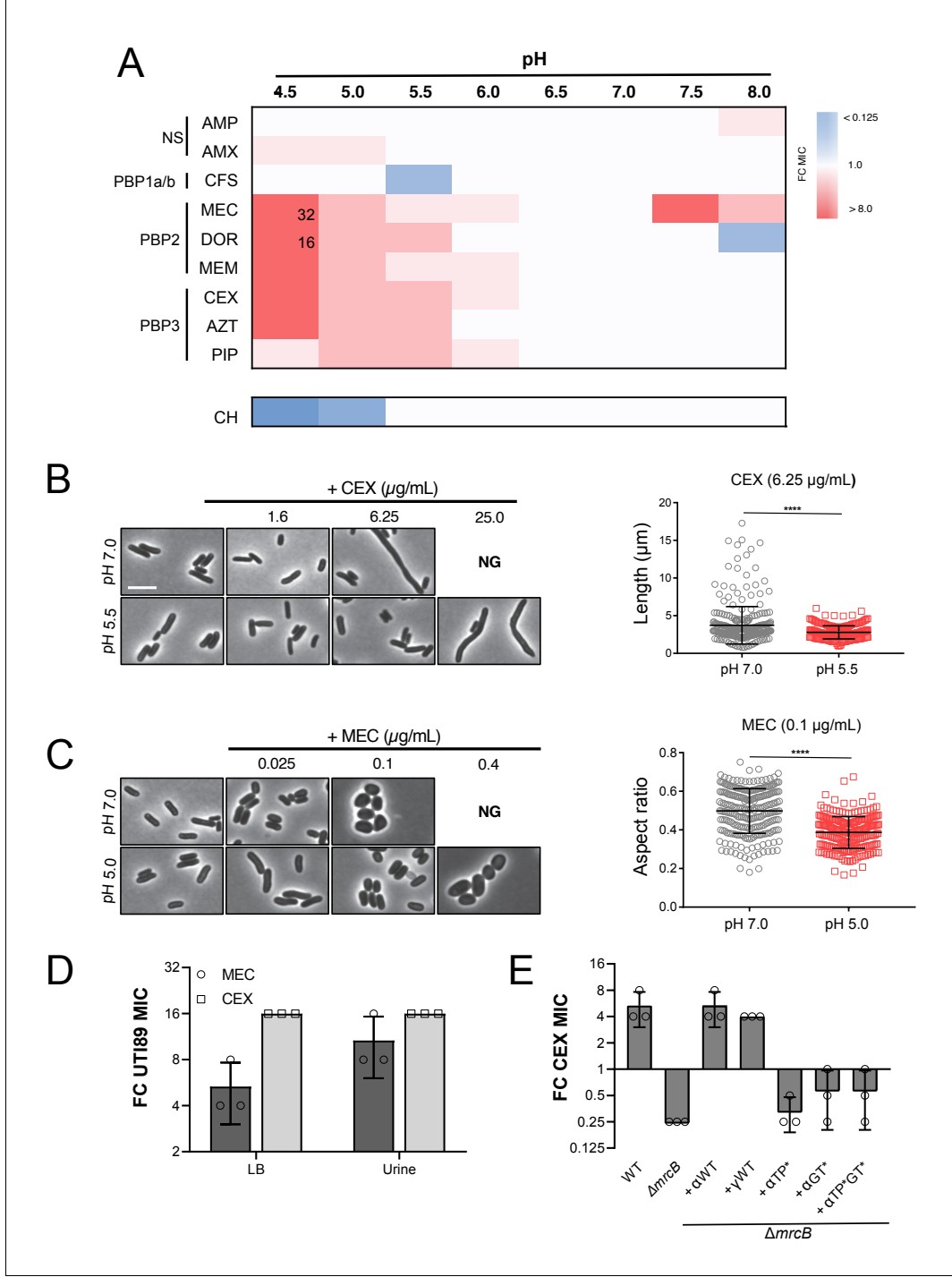

**Figure 6.** Intrinsic resistance to PBP2 and PBP3-targeting β-lactams at low pH. (**A**) Heat map summarizing fold change in minimum inhibitory concentrations (MIC) of antibiotics for strain MG1655 cultured in LB (pH 4.5–8.0) after 20 hr. Cells in heat map are colored based on median fold change (FC) in MIC at indicated pH compared to pH 7.0 from at least three biological replicates. Fold change values of >8 are indicated in black inside relevant cell. Untransformed median MIC values can be viewed in ***Supplementary file 4***. Abbreviations for antibiotic names are as follows: AMP, ampicillin; AMX, amoxicillin; CFS, cefsulodin; MEC, mecillinam; DOR, doripenem; MEM, meropenem; CEX, cephalexin; AZT, aztreonam; PIP, piperacillin; CH, chloramphenicol. Predominant cellular PBP target is indicated to the left. (**B**) Representative micrographs of cells treated with PBP3 inhibitor cephalexin (CEX) at pH 7.0 and pH 5.5. Distribution of cell lengths at sub-MIC concentration at pH 7.0 and 5.5 (n = 303 and 250) are shown to the right. (**B**) Representative micrographs of cells treated with PBP2 inhibitor mecillinam (MEC) cultured

*Figure 6 continued on next page*

*Figure 6 continued*

at pH 7.0 or pH 5.0. Distribution of cell aspect ratios (width/length) at sub-MIC concentration at pH 7.0 and 5.0 (n = 250 and 250) are shown to the right. For both panels A and B, scale bar indicates 3 μm, and NG denotes 'no growth' observed at the indicated concentration of antibiotic. Error bars represent SD. Significance was assessed by a Kruskal-Wallis test with asterisks denoting significance as follows: ****, p<0.0001. (**D**) Fold change in minimum inhibitory concentration of *E. coli* strain UTI89 to cephalexin (CEX) and mecillinam (MEC) grown at pH 5.0 compared to pH 7.0 in broth culture and in urine. Untransformed MIC values can be viewed in **Supplementary file 4**. (**E**) Fold change in minimum inhibitory concentration to cephalexin (CEX) for indicated strains grown at pH 5.5 compared to pH 7.0. EAM696 (Δ*mrcB*) derivatives producing PBP1b variants were grown in the presence of 10 μM IPTG. Untransformed MIC values can be viewed in **Supplementary file 5**. Bars represent mean fold change in minimum inhibitory concentration ± SD across at least three biological replicates.

DOI: https://doi.org/10.7554/eLife.40754.022

The following figure supplements are available for figure 6:

**Figure supplement 1.** Stability of β-lactam antibiotics across pH values.
DOI: https://doi.org/10.7554/eLife.40754.023

**Figure supplement 2.** Mecillinam resistant cells at pH 8.0 remain rounded.
DOI: https://doi.org/10.7554/eLife.40754.024

**Figure supplement 3.** Proton motive force, AmpC β-lactamase, and outer membrane permeability do not confer pH-dependent resistance to cephalexin.
DOI: https://doi.org/10.7554/eLife.40754.025

**Figure supplement 4.** Δ*mrcB* abolishes low pH-dependent resistance independent of growth rate and β-lactam sensitivity.
DOI: https://doi.org/10.7554/eLife.40754.026

(**Müller et al., 2007**; **Bertsche et al., 2006**; **Banzhaf et al., 2012**). Our current study was limited to interrogating the pH specialization of nonessential cell wall enzymes. Future work investigating the influence of pH on the essential components of the elongation and division machinery is thus necessary to determine whether the activity of these complexes affects the cell's class A PBP preference.

Although PBP1a and PBP1b share an essential role in growth and cell wall integrity under standard culture conditions (i.e. nutrient rich, neutral pH growth medium) (**Suzuki et al., 1978**), there had been previous hints these enzymes were not interchangeable. As previously mentioned, each class A PBP possess unique interaction partners and subcellular localization profiles (**Müller et al., 2007**; **Bertsche et al., 2006**; **Banzhaf et al., 2012**; **Gray et al., 2015**; **Leclercq et al., 2017**). Mutants defective for each enzyme also display differential susceptibility to antibiotic treatment (discussed below), osmotic shifts, and mechanical stress (**Yousif et al., 1985**; **Paradis-Bleau et al., 2014**; **Auer et al., 2016**) and have different roles in de novo regeneration of rod shape (**Ranjit et al., 2017**). We anticipate further study of these synthases—as with other 'redundant' cell wall autolysins—under nonstandard culture conditions will continue to reveal unique roles for these enzymes in cell wall biogenesis.

## Plasticity in cell wall metabolism influences intrinsic resistance to cell wall active antibiotics

Analogous to alternative PBP usage in methicillin resistant *Staphylococcus aureus* (**Berger-Bächi, 1999**; **Chan et al., 2016**), our results suggest that environment-driven plasticity in *E. coli* PG synthesis may have consequences for intrinsic resistance to β-lactam antibiotics with a narrow target specificity (**Figure 6**). Strikingly, we find that culturing *E. coli* in low pH media is sufficient increase intrinsic resistance to PBP2 and PBP3-targeting β-lactams up to 32-fold. Considering *E. coli* encounters a wide range of pH environments across host niches (**Watson et al., 1972**; **Bilobrov et al., 1990**), our observation reinforces the importance of conducting antibiotic susceptibility testing under physiologically relevant conditions (**Ersoy et al., 2017**; **Thulin et al., 2017**).

Mechanistically, our data demonstrate low pH-dependent resistance specifically requires PBP1b activity. PBP1b has previously been implicated in intrinsic resistance to β-lactam antibiotics: cells defective for PBP1b or LpoB are hypersensitive to a wide variety of β-lactams (**Nichols et al., 2011**; **Paradis-Bleau et al., 2014**; **García del Portillo and de Pedro, 1990**). Likewise, elevated PBP1b activity protects cells from the lethality of the PBP2 specific antibiotic mecillinam (**Lai et al., 2017**).

At present, the precise role for PBP1b in β-lactam protection remains unclear. It appears to be a function specifically endowed to PBP1b, for cells defective for PBP1a production do not display differential β-lactam susceptibility (*Nichols et al., 2011*; *García del Portillo and de Pedro, 1990*), highlighting another possible source of specialization between the class A PBPs. Interestingly, we and others have observed cells cultured in acidic media exhibit near normal morphology in the presence of concentrations of PBP2 or PBP3 inhibitors that lead to cell rounding or filamentation at neutral pH (*Figure 6B,C*) (*Goodell et al., 1976*). We thus speculate that PBP1b may play a role in preserving the normal functions of the elongation and division machinery in low pH environments. PBP1b may substitute for the essential function of PBP2 and PBP3 in these complexes (*Modell et al., 2014*) or may indirectly support growth by serving a quality control function (*Morè et al., 2019*). Clarifying PBP1b's role in intrinsic resistance to β-lactams will shed light on the mechanics of cell wall biogenesis and inform the design and use of novel antibiotic therapies.

### Apparent redundancy ensures fitness across environmental conditions

Apart from influencing antibiotic resistance, environmental specialization of cell wall enzymes is likely a key adaptation that allows *E. coli* to thrive across an unusually wide pH range (pH ~4–9) and even tolerate extreme pH shocks, such as during transient exposure to gastric acid (pH ~2) (*Jordan et al., 1999*). Plasticity in the cell wall synthesis machinery likely works in concert with the organism's ability to modify extracellular pH through the export of acidic and alkaline substrates (*Lu et al., 2013*; *Krulwich et al., 2011*). In this context, pH-specialist cell wall enzymes may function in part to maintain cell wall integrity until the extracellular media reaches a growth-permissive pH.

Among other pH tolerant organisms, strategies employed to expand growth across wide pH ranges are likely to vary, even between closely related species. *S. enterica* serovar Typhimurium encodes a PBP3 paralog, termed $PBP3_{SAL}$, that is active during growth in acidic environments, including intracellularly in the phagolysosome (*Castanheira et al., 2017*). As $PBP3_{SAL}$ is restricted to *Salmonella*, *Enterobacter*, and *Citrobacter* spp., alternative mechanisms to cope with changing pH environments must exist. Elucidating the requirements for pH-dependent growth in organisms outside the *Enterobacteriaceae* will shed light on whether class A PBPs, which are broadly conserved across bacteria (*Typas et al., 2010*), play a central role in the process.

Apart from pH, we anticipate enzyme specialists exist across environmental conditions; redundancy in cell wall enzymes is present throughout bacterial species, even among those that only grow at a narrow pH range (*Pazos et al., 2017*). *B. subtilis*, for example, encodes 16 PBPs, yet its growth is restricted to pH 6.0–9.0 (*Wilks et al., 2009*). Ionic strength, osmolality, and temperature also vary across the diverse habitats bacteria occupy. Like pH, these factors may have significant impacts on the chemical and physical properties of the periplasm and the extracytoplasmic space of Gram-positive bacteria. In support of this idea, PBP2 from *Caulobacter crescentus* displays differential localization patterns as a function of extracellular osmolality (*Hocking et al., 2012*), and the lytic transglycosylase MltA from *E. coli* is ~10 times more active at 30°C than at 37°C in vitro (*van Straaten et al., 2005*; *Lommatzsch et al., 1997*). We expect environmental specialization may also underlie the high levels of redundancy in other periplasmic protein classes, including sugar transporters, efflux pump adapter proteins, and chaperones (*Jensen et al., 2002*; *Smith and Blair, 2014*; *Rizzitello et al., 2001*). Nevertheless, it is clear that improved understanding of the contribution of many enzymes to bacterial fitness in nature demands a departure from standard growth conditions used to study bacterial physiology in the lab.

## Materials and methods

**Key resources table**

| Reagent type (species) or resource | Designation | Source or reference | Identifiers | Additional information |
|---|---|---|---|---|
| Gene (*Escherichia coli*) | *mrcB* | NA | EcoCyc:EG10605 | |
| Gene (*E. coli*) | *mrcA* | NA | EcoCyc:EG10748 | |
| Gene (*E. coli*) | *lpoA* | NA | EcoCyc:G7642 | |

*Continued on next page*

*Continued*

| Reagent type (species) or resource | Designation | Source or reference | Identifiers | Additional information |
|---|---|---|---|---|
| Gene (*E. coli*) | *lpoB* | NA | EcoCyc:G6565 | |
| Strain, strain background (*E. coli*) | MG1655; wild-type | PMID:6271456 | | |
| Strain, strain background (*E. coli*) | UTI89 | PMID:11402001 | | |
| Antibody | anti-PBP1a | Gift of Waldemar Vollmer | | (1:5000) |
| Antibody | anti-PBP1b | Gift of Waldemar Vollmer | | (1:1000) |
| Antibody | anti-FtsZ | Gift of David Weiss | | (1:5000) |
| Recombinant DNA reagent | *mrcB::kan; ΔmrcB* | PMID:16738554; Coli Genetic Stock Center | CGSC:JW0145-1 | |
| Recombinant DNA reagent | *mrcA::kan; ΔmrcA* | PMID:16738554; Coli Genetic Stock Center | CGSC:JW3359-1 | |
| Recombinant DNA reagent | *lpoA::kan; ΔlpoA* | PMID:16738554; Coli Genetic Stock Center | CGSC:JW3116-1 | |
| Recombinant DNA reagent | *lpoB::kan; ΔlpoB* | PMID:16738554; Coli Genetic Stock Center | CGSC:JW5157-1 | |
| Recombinant DNA reagent | pCP20 | PMID:10829079 | | |
| Recombinant DNA reagent | pUM1Bα | PMID:12949085 | | |
| Recombinant DNA reagent | pUM1Bγ | PMID:12949085 | | |
| Recombinant DNA reagent | pUM1Bα* | PMID:12949085 | | |
| Recombinant DNA reagent | pUM1BTG*α | PMID:12949085 | | |
| Recombinant DNA reagent | pUM1BTG*α* | PMID:12949085 | | |
| Chemical compound, drug | Ampicillin Sodium Salt | Sigma Aldrich | Catalog:A9518 | |
| Chemical compound, drug | Amoxicillin | Sigma Aldrich | Catalog:A8523 | |
| Chemical compound, drug | Cefsulodin Sodium Salt Hydrate | Sigma Aldrich | Catalog:C8145 | |
| Chemical compound, drug | Mecillinam | Sigma Aldrich | Catalog:33447 | |
| Chemical compound, drug | Doripenem Hydrate | Sigma Aldrich | Catalog:SML1220 | |
| Chemical compound, drug | Meropenem Hydrate | Sigma Aldrich | Catalog:M2574 | |
| Chemical compound, drug | Cephalexin | Sigma Aldrich | Catalog:33989 | |
| Chemical compound, drug | Aztreonam | Sigma Aldrich | Catalog:A6848 | |
| Chemical compound, drug | Piperacillin Sodium Salt | Sigma Aldrich | Catalog:P8396 | |
| Software, algorithm | SuperSegger | PMID:27569113 | | |
| Software, algorithm | FIJI | PMID:22743772 | | |
| Other | Propidium iodide | Sigma Aldrich | Catalog:81845 | |

## Bacterial strains, plasmids, and growth conditions

Unless otherwise indicated, all chemicals, media components, and antibiotics were purchased from Sigma Aldrich (St. Louis, MO). Bacterial strains and plasmids used in this study are listed in *Supplementary file 1* and *Supplementary file 2*, respectively. All deletion alleles were originally provided by the Coli Genetic Stock (*Baba et al., 2006*) and transduced into *E. coli* strain MG1655. For the hits identified in the growth rate screen, the expected mutation was confirmed by diagnostic PCR with *Taq* polymerase. Unless otherwise indicated, strains were grown in lysogeny broth (LB) media (1% tryptone, 1% NaCl, 0.5% yeast extract) supplemented with 1:10 MMT buffer (1:2:2 molar ratio of D,L-malic acid, MES, and Tris base) to vary media pH values between pH 4–9. AB defined media (*Clark and Maaløe, 1967*) was fixed to indicated pH values with addition of 5M HCl or 5M NaOH. Uropathogenic *E. coli* strain UTI89 was cultured in urine provided by a healthy donor and supplemented with MMT buffer to fix the pH. When selection was necessary, cultures were

supplemented with 50 µg/mL kanamycin (Kan) and 25–100 µg/mL ampicillin (Amp). Cells were grown at 37°C either in 96-well microtiter plates shaking at 567 cpm or in glass culture tubes shaking at 200 rpm for aeration.

## Growth rate measurements

Strains were grown from single colonies in glass culture tubes in LB +MMT buffer (pH 6.9) to mid-log phase ($OD_{600}$ ~0.2–0.6), pelleted, and re-suspended to an $OD_{600}$ of 1.0 (~$1 \times 10^9$ CFU/mL). Cells were diluted and inoculated into fresh LB +MMT buffer at various pH values in 96-well plates (150 µl final volume) at $1 \times 10^3$ CFU/mL. Uncovered plates sealed with gas permeable membrane strips were grown at 37°C shaking for 20 hr in a BioTek Eon microtiter plate reader, measuring the $OD_{600}$ of each well every ten minutes. Mass doublings per hour (DPH) was calculated by least squares fitting of early exponential growth ($OD_{600}$ 0.005–0.1) in R. Best fit lines with an $R^2$ value below 0.95 were excluded from further analysis. Examples of growth curves and fit lines are presented in *Figure 1—figure supplement 1*, along with a sample script (*Supplementary file 7*) and representative source data (*Figure 1—source datas 1–3*). To allow for direct comparison between wild type and mutant growth, some panels present % wild type growth, which represents the $DPH_{Mutant}/DPH_{WT} \times 100$.

## Microscopy and time lapse imaging

For time lapse imaging experiments, cells were grown from a single colony in LB +MMT buffer (pH 6.9) to early exponential phase ($OD_{600}$ ~0.05–0.1) then mounted onto 1.0% agarose pads at pH 4.5, 6.9, or 8.0. Where indicated, propidium iodide was added to the agarose pad at a final concentration of 1.5 µM. Cells were allowed to dry on pads 10 min prior to imaging. All phase contrast and fluorescence images were acquired on a Nikon Ti-E inverted microscope (Nikon Instruments, Inc) equipped with a 100X Plan N (N.A. = 1.45) Ph3 objective, X-Cite 120 LED light source (Lumen Dynamics), and an OrcaERG CCD camera (Hammamatsu Photonics, Bridgewater, N.J.). Filter sets were purchased from Chroma Technology Corporation. The objective was pre-heated to 37°C using an objective heater. Image capture and analysis was performed using Nikon Elements Advanced Research software. Cell death quantification was determined by cells uniformly stained with propidium iodide, and terminal lytic phenotype of cells was determined by assessment of the frames immediately preceding propidium iodide incorporation. Single cell elongation rate, defined as k = ΔL/Δt/L, was determined in the MATLAB-based program SuperSegger (*Stylianidou et al., 2016*). Cells that lysed during the movie, indicated by negative k values, were filtered out prior to analysis.

For class A PBP localization studies, cells were grown in AB minimal media supplemented with 0.2% maltose and 250 µM IPTG overnight then sub-cultured into fresh media the following morning. Cells were grown to $OD_{600}$ 0.1–0.2 at 37°C then fixed by adding 20 µL of 1M NaPO4, pH 7.4, and 100 µL of fixative (fixative = 1 mL 16% paraformaldehyde + 6.25 µL 8% glutaraldehyde). Samples were incubated at room temperature for 15 min, then on ice for 30 min. Fixed cells were pelleted, washed three times in 1 mL 1X PBS, pH 7.4, then resuspended in GTE buffer (glucose-tris-EDTA) and stored at 4°C. Quantification shown in *Figure 5—figure supplement 2* was performed in FIJI (*Schindelin et al., 2012*). Briefly, an intensity profile was generated for each cell by drawing a line across the midline of the cell from pole to pole. Maximum length and intensity were normalized to 100%. Cells enriched for class A PBP localization at the cell periphery would be expected to have increased intensity at the cell poles (0% and 100% of cell length).

## Scanning electron microscopy

Wild type and ∆mrcB cells were grown to mid-exponential phase in MMT buffered pH 6.9 LB media and back-diluted to an $OD_{600}$ = 0.1 into either pH 6.9 or 4.5 media. Cells were allowed to grow for an additional hour, fixed as described above, and applied to poly-lysine coated coverslips. Post fixation, samples were rinsed in PBS 3 times for 10 min each followed by a secondary fixation in 1% $OsO_4$ in PBS for 60 min in the dark. The coverslips were then rinsed three times in ultrapure water for 10 min each and dehydrated in a graded ethanol series (50%, 70%, 90%, 100% x2) for 10 min each step. Once dehydrated, coverslips were then loaded into a critical point drier (Leica EM CPD 300, Vienna, Austria) which was set to perform 12 $CO_2$ exchanges at the slowest speed. Once dried, coverslips were then mounted on aluminum stubs with carbon adhesive tabs and sputter coated

with 6 nm of iridium (Leica ACE 600, Vienna, Austria). After coating, the samples were then loaded into a FE-SEM (Zeiss Merlin, Oberkochen, Germany) imaged at 3 KeV with a probe current of 178 pA using the Everhart Thornley secondary electron detector.

## SDS-PAGE and immunoblotting

Strains were grown from a single colony in LB + 1:10 MMT at pH 4.8, 6.9, or 8.2 to mid-log phase (OD$_{600}$ ~0.2–0.6), back-diluted to 0.005 in 5 mL of media, and grown to an OD$_{600}$ between 0.2–0.3. Samples were pelleted, re-suspended in 2x Laemmli buffer to an OD$_{600}$ ~20, and boiled for ten minutes. Samples (10 µl) were separated on 12% SDS-PAGE gels by standard electrophoresis and transferred to nitrocellulose membranes. Blots were probed with PBP1b (1:1000), PBP1a (1:5000), and FtsZ rabbit antisera (1:5000) and HRP-conjugated secondary antibody (1:2000-1:10000; goat α-rabbit). Blots were imaged on a LiCor Odyssey imager. Quantitation was determined in ImageJ and normalized to FtsZ levels as an internal loading control.

## In vitro protein materials and interaction and activity assays

Lipid II versions were prepared as previously described (*Bertsche et al., 2005*; *Breukink et al., 2003*). The following proteins were prepared as previously described; PBP1B (*Bertsche et al., 2006*), LpoB (*Egan et al., 2014*), PBP1A (*Born et al., 2006*), and LpoA (*Jean et al., 2014*). Antisera against PBP1A and PBP1B were obtained from Eurogentec (Liege, Belgium) and purified over an antigen column as described previously (*Bertsche et al., 2006*).

SPR experiments were performed as previously described (*Egan et al., 2014*). LpoA and LpoB samples were prepared for injection over the PBP surface by 1:1 serial dilution from 10 µM to 19.5 nM. Assays were performed in triplicate at 25 °C, at a flow rate of 75 µL/min and with an injection time of 5 min. The running buffers consisted of 20 mM of either; sodium acetate pH 4.8, HEPES/NaOH pH 6.9, or Tris/HCl pH 8.2 with 150 mM NaCl, and 0.05% Triton X-100. The dissociation constant (K$_D$) was calculated by non-linear regression using SigmaPlot 13 software (Systat Software Inc). Continuous fluorescence GTase assays were performed as described previously (*Egan and Vollmer, 2016*) with modification. Final buffer composition was 20 mM of either; sodium acetate pH 4.8, HEPES/NaOH pH 6.9, or Tris/HCl pH 8.2 plus 150 mM NaCl, 10 mM MgCl$_2$, and 0.05% Triton X-100. Enzymes were assayed alone at 1 µM at 37°C, and at 0.2 µM in the presence of 0.5 µM regulator(s) at 25°C. The muramidase usually included in the assay samples to digest newly synthesized glycans, thereby improving fluorescence signal, was omitted to avoid indirect pH effects on observations. The slopes of the resulting curves correlate with the GTase rate and were calculated at their fastest point using linear regression in Excel 2016 (Microsoft). When presented in *Figure 5—figure supplement 3*, the values are inverted from negative for simplicity. Measurement of total PG synthesis activity using radiolabelled lipid II substrate was performed as previously described (*Biboy et al., 2013*) using enzyme (0.5 µM) and regulator(s) (2 µM) in the same three buffers indicated for the GTase assay. Total TPase activity was calculated as the percentage of muropeptide products known to be produced by this domain's function, including peptide cross-linking and DD-carboxypeptidase activity.

## Antibiotic susceptibility testing

For determination of minimum inhibitory concentrations, cells were grown from a single colony in LB media at the indicated pH to mid-exponential phase (OD$_{600}$ ~0.2–0.6) at 37°C with aeration and then inoculated at 1 × 10$^5$ CFU/mL into LB media of the same pH in sterile 96-well plates with a range of two-fold dilutions of the indicated antimicrobial agent (final volume, 150 µL). Plates were incubated at 37°C shaking for 20 hr before determination of the well with the lowest concentration of the antibiotic that had prevented growth by visual inspection.

## Antibiotic stability testing

Antibiotics were incubated for 20 hr in LB media at pH 4.5, 7.0, or 8.0 and then diluted into 96-well plates containing LB media (pH 7) and 1 × 10$^5$ CFU/mL MG1655. Plates were then incubated at 37°C shaking for 20 hr before determination of the compound's minimum inhibitory concentration.

## Terminal phenotype assessment

Cells from minimum inhibitory concentration assays were spotted (5 µL) onto 1.0% agarose pads 20 hr post-treatment and imaged by phase contrast microscopy to track cell morphology in response to antibiotic treatment across pH values. Growth rate was monitored by $OD_{600}$ in the BioTek Eon plate reader to confirm all cells examined were in the same growth phase and at approximately the same optical density prior to imaging. Cell dimensions were quantified in the MATLAB-based program SuperSegger (*Stylianidou et al., 2016*).

## Quantification and statistical analysis

A minimum of three biological replicates were performed for each experimental condition unless otherwise indicated. Data are expressed as means ± standard deviation (SD) or standard error of the mean. Statistical tests employed are indicated in the text and corresponding figure legend. Analysis was performed in R or GraphPad Prism. Asterisks indicate significance as follows: *, $p < 0.05$; **, $p < 0.01$; ***, $p < 0.001$; ****, $p < 0.0001$.

## Acknowledgements

We thank Tom Bernhardt and Joe Vogel for gifts of strains and plasmids, respectively. We appreciate sample preparation and electron microscopic imaging assistance from Matthew Joens, Daniel Geanon, Greg Strout and Dr. James Fitzpatrick from the Washington University Center for Cellular Imaging which is supported by Washington University School of Medicine, The Children's Discovery Institute of Washington University and St. Louis Children's Hospital (CDI-CORE-2015–505) and the Foundation for Barnes-Jewish Hospital (3770). We are indebted to members of the Levin and Zaher labs for fruitful discussions on technical and philosophical matters related to this this research, as well as Corey Westfall, Joseph Merriman, and Katharina Peters for critical reading of this manuscript.

## Additional information

### Funding

| Funder | Grant reference number | Author |
| --- | --- | --- |
| National Science Foundation | DGE-1745038 | Elizabeth A Mueller |
| Wellcome | 101824/Z/13/Z | Waldemar Vollmer |
| National Institutes of Health | GM127331 | Petra Anne Levin |
| National Institutes of Health | GM64671 | Petra Anne Levin |

The funders had no role in study design, data collection and interpretation, or the decision to submit the work for publication.

### Author contributions

Elizabeth A Mueller, Conceptualization, Formal analysis, Funding acquisition, Validation, Investigation, Visualization, Methodology, Writing—original draft, Writing—review and editing; Alexander JF Egan, Conceptualization, Formal analysis, Investigation, Visualization, Methodology, Writing—review and editing; Eefjan Breukink, Resources, Writing—review and editing; Waldemar Vollmer, Petra Anne Levin, Conceptualization, Resources, Supervision, Funding acquisition, Project administration, Writing—review and editing

### Author ORCIDs

Elizabeth A Mueller (iD) http://orcid.org/0000-0001-5482-6551
Petra Anne Levin (iD) http://orcid.org/0000-0003-2071-0547

### Decision letter and Author response

Decision letter https://doi.org/10.7554/eLife.40754.036
Author response https://doi.org/10.7554/eLife.40754.037

## Additional files

### Supplementary files

• Supplementary file 1. Bacterial strains used in this study.
DOI: https://doi.org/10.7554/eLife.40754.027

• Supplementary file 2. Plasmids used in this study.
DOI: https://doi.org/10.7554/eLife.40754.028

• Supplementary file 3. Summary of growth rate screen. Supports *Figure 1*. Presents mean mass doubling time ± standard deviation of each cell wall mutant at pH 4.8, 6.9, and 8.2 during preliminary screen (n = 3).
DOI: https://doi.org/10.7554/eLife.40754.029

• Supplementary file 4. β-lactam sensitivity of MG1655 across pH conditions. Supports *Figure 6A*. Presents median minimum inhibitory concentrations of indicated β-lactam antibiotics to MG1655 across pH conditions of at least three biological replicates. Values are represented as µg/mL.
DOI: https://doi.org/10.7554/eLife.40754.030

• Supplementary file 5. β-lactam sensitivity of UTI89 across pH conditions. Supports *Figure 6D*. Presents median minimum inhibitory concentrations of cephalexin (CEX) and mecillinam (MEC) to UTI89 across pH conditions in LB and in urine (n = 3). Values are represented as µg/mL.
DOI: https://doi.org/10.7554/eLife.40754.031

• Supplementary file 6. Susceptibility of strains producing PBP1b variants to cephalexin across pH conditions. Supports *Figure 6E*. Presents median minimum inhibitory concentrations of cephalexin to MG1655 and PBP1b derivatives across pH conditions (n = 3). Values are represented as µg/mL.
DOI: https://doi.org/10.7554/eLife.40754.032

• Supplementary file 7. Representative script used to analyze bacterial growth rate datasets. Supports *Figure 1* and *Figure 1—figure supplement 1*. This sample script uses source data from *Figure 1—source data 2*.
DOI: https://doi.org/10.7554/eLife.40754.033

• Transparent reporting form
DOI: https://doi.org/10.7554/eLife.40754.034

### Data availability

All data generated or analyzed during this study are included in the manuscript and supporting files.

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
