## [Decision Letter]

[**Editorial note:** This article has been through an editorial process in which the authors decide how to respond to the issues raised during peer review. The Reviewing Editor's assessment is that all the issues have been addressed.]

Evaluation of resubmission:

The authors have now responded to each of the issues raised by the three reviewers. In general, the responses are thorough and appropriate. The changes made to the text as well as the additional experiments performed have now strengthened many of the conclusions and claims made in the paper. In particular, the evidence supporting the pH-specialization of PBP1a and PBP1b has been improved. This includes new in vitro studies, done collaboratively with the Vollmer group, showing that PBP1a activity is significantly reduced at pH 4.8, potentially explaining why PBP1b activity becomes required at lower pH. Precisely how pH affects PBP1a activity still is not totally clear - an attempt was made to measure stimulation by LpoA but technical difficulties prevented a clear assessment of this. The authors have also addressed a key concern from one reviewer regarding the increased resistance to a sub-class of beta-lactam antibiotics at low pH showing that the effects are likely not due to non-specific changes in permeability. All told, the changes made to the paper have improved it and strengthened the major conclusions.

Decision letter after peer review:

Thank you for submitting your article "Plasticity of *E. coli* cell wall metabolism promotes fitness and antibiotic resistance across environmental conditions" for consideration by *eLife*. Your article has been reviewed by three peer reviewers, including Michael T Laub as the Reviewing Editor and Reviewer #1, and the evaluation has been overseen by Gisela Storz as the Senior Editor.

The Reviewing Editor has highlighted the concerns that require revision and/or responses, and we have included the separate reviews below for your consideration. If you have any questions, please do not hesitate to contact us.

Summary of major concerns:

This paper presents data to support the intriguing conclusion that the apparent redundancy of cell wall synthetic proteins may stem from their specialization to different environments. The reviewers agreed that the paper was well-written and contained a substantial body of data to demonstrate that PBP1b, in particular, assumes a pivotal role in low pH environments. There were, however, a number of major concerns raised, detailed in the individual reviews below. Most notably, the reviewers were concerned about whether PBP1a is really "specialized" for alkaline pH and hence whether PBP1a/b redundancy has evolved to ensure optimal activities in opposing pH niches. They also thought the paper would benefit from additional data and insights into the notion that PBP1b may replace PBP2/3 in the elongation and division machineries at low pH. There were also concerns about the statistics and number of replicates in several figures.

You can see below that reviewer 2 initially was quite positive. However to be complete, we want to give you a sense of the subsequent discussion where that reviewer stated:

"I was obviously more enthusiastic about the story than both of you, although I too was surprised to see it in *eLife*. I think that you both brought up important points and that my liking of the overarching idea of specialization made me miss some of the issues."

"So, thinking about everyone's comments and my new perspective, I believe that the paper is not at par for *eLife*. It presents a very interesting model to explain the so-called redundancy and lack of phenotypes. I like the idea that the lack of phenotypes reflects that we are looking under the wrong conditions (which the paper illustrates) because those enzymes have been selected for in the real world for billions of years. I also like the warning about antibiotic sensitivity being affected by environmental conditions like pH. However, the problem is that the authors overinterpret some of their results. It seems that the story boils down to PBP1A specifically not being able to work at low pH for an unknown mechanism. PBP1B seems to be able to work under all conditions, although slightly worse than 1A at high pH, but the data do not support the perfect picture of Figure 7. They could get more data in response to reviews but, most importantly, they will have to revise the model."

Separate reviews (please respond to each point):

*Reviewer #1:*

This paper examines the pH-dependencies of PBP1a and PBP1b, arguing that each is "specialized" for a different pH. The data are relatively clean and soundly interpreted in most cases. Overall, the paper paints a compelling picture that the apparent redundancy of cell wall enzymes may reflect specialization for different environments. Below are several issues that I think the authors should respond to before publication – in some cases, this could involve significant new experiments, but I think such experiments would substantially add to the paper's mechanistic depth and long-term impact on the field.

1) Figure 5C shows PBP1b levels – but what about PBP1a?

2) Is there any significance to the clustering of GFP-PBP1a at low pH? I was sort of surprised that this observation wasn't followed up at all.

3) PBP1b provides resistance to various antibiotics at low pH. But I don't really understand why. If the model is that PBP1a is more active and neutral and alkaline pH with PBP1b more active at low pH, why isn't PBP1a leading to comparable levels of resistance at the higher pHs? Do PBP1b and PBP1a, in fact, have different activities? I thought they were carrying out the same reaction, with one more active at low pH and the other more active at higher pHs, so the source of the difference in antibiotic susceptibility doesn't really make sense to me, but maybe I'm missing something here.

4) I really think it's a lost opportunity to not provide more insight into the molecular basis for the pH differences in PBP1a and PBP1b. It appears that PBP1b activity can be reconstituted in vitro, including with LpoB and FtsN, so testing things in vitro at different pHs so testing things in vitro at different pHs seems potentially feasible. I would encourage the authors to at least consider whether something in this regard can be done to strengthen and enhance the paper.

5) I wondered whether there are phylogenetic analyses to support the notion that PBP1a and PBP1b have evolved, at least in part, to allow cells to cope with different pH conditions. Are there strains of *E. coli* lacking PBP1b or even closely related species lacking it? If so, can they not tolerate low pHs and/or would their PBPs complement a PBP1a/1b double deletion only at certain pHs?

*Minor Comments:*

Abstract, last sentence: Awkward sentence – rephrase for clarity.

"providing a force" maybe not "providing the force" in light of recent findings from KC Huang et al.

"in in" typo.

Introduction, last sentence: Sort of a cryptic end to the Introduction – clarify why this is concerning and what the consequences are.

Subsection “Identification of pH specialist cell wall synthases and hydrolases”, first paragraph: The authors probably need to do an ANOVA test to control for multiple hypothesis testing. Even with a p < 0.01 to increase stringency, I think an ANOVA is the right choice here.

What does 'cytoskeletal-independent' mean? I don't think of either MreB or FtsZ as cytoskeletal elements anymore – reword for accuracy/precision.

"optimized" is a strong word…maybe "tuned" would be better. I just always think it's tough to show something is truly optimal.

Subsection “aPBP activity promotes cell wall integrity across pH environment”, first paragraph: typo, "cells" not "cell".

Typo "previous a proteomic".

*Reviewer #2:*

Most bacteria build a peptidoglycan cell wall that protects them from osmotic lysis. While enzymes required for the synthesis of the peptidoglycan precursor (lipid-linked disaccharide pentapeptide) are essential, many of the enzymes participating in the construction of the cell wall are not. In *Escherichia coli*, there is functional redundancy between two class a penicillin-binding proteins (PBPs) since removal of both PBP1a and PBP1b causes lethality. This has led to the idea that functional redundancy might be widespread among enzymes that synthesize and modify the cell wall, since removal of individual factors does not typically confer phenotypes. Here, Mueller and Levin challenge the view that these enzymes are simply redundant. They showed that the so-called "redundant" PBP1a and PBP1b enzymes have each specialized to function in different pH ranges. While PBP1a is dispensable in acidic conditions, PBP1b is essential; in contrast, in alkaline environments, PBP1b is dispensable, but PBP1a is required for optimal survival. This is a significant discovery and it is possible that, as the authors propose, similar specialization might apply to other enzymes involved in peptidoglycan synthesis and remodeling. Furthermore, the authors demonstrate that the intrinsic resistance to antibiotics can be different based on environmental pH.

The manuscript is well written, experiments were well done, conclusions are justified, and the work reports significant findings.

There is only one issue that should be easily addressed to strengthen one of the claims:

The authors used a propidium iodide assay to quantify lysis. This assay measures the uptake of the dye not necessary lysis. Although the images of cells in Figure 3 and Figure 4 show that some cells have lysed, the authors should track loss of cytoplasmic contents (e.g. leakage of cytoplasmic GFP or LacZ) to demonstrate and quantify lysis. It would also be nice to complement lysis data with a kill curve (determine number of cells alive by CFUs after exposure to the specific pH). The authors could still leave the propidium iodide data to reflect envelope problems (but not necessarily lysis or even death).

*Minor Comments:*

1) Delete repeated "in".

2) Subsection “Identification of pH specialist cell wall synthases and hydrolases”, second paragraph: I think it should state "six" mutants. Not "seven".

3) Subsection “aPBP activity ensures fitness across a wide pH range”, first paragraph: Denome et al., 1999, might not be the best reference to cite the synthetic lethality of PBP1a and 1b because the pbp1a mutant used in that study contained additional mutations, as described in Meberg et al., J Bacteriol. 183:6148-9; PMID: 11567017. It would be more appropriate to cite the first report of the synthetic lethality: Suzuki et al. (1978) PNAS 75: 664-668; https://doi.org/10.1073/pnas.75.2.664.

*Reviewer #3:*

Bacteria maintain a tremendous enzymatic redundancy when it comes to building up their cell wall. In this paper, the authors systematically assess the contribution of 33 enzymes (carrying 7 distinct enzymatic functions) to the growth rate of *E. coli* in low and high pH, both commonly encountered environments for *E. coli* in the human body. The authors then focus on one group, the bifunctional synthases, and provide evidence that its 2 main members, PBP1a and PBP1b are specialized for acting in high and low pH, respectively. They also link this PBP1b specificity to increased resistance to a subgroup of β-lactam inhibitors at acidic environments. Overall, the authors use this example to propose that division of labor and environment-specific action is the main reason behind the apparent redundancy of cell wall enzymes.

The strengths of this paper are that:

• it is clearly and well written;

• it systematically interrogates the relationships between a specific environment (pH) and cell wall biosynthetic enzymes;

• it makes a stronger case for environment-dependent changes in the activity of cell-wall enzymes;

• it provides strong evidence for the crucial role of PBP1b at low pH (which I think the authors should focus on);

• it provides preliminary evidence that there may be a striking reorganization of the division and elongation machineries at low pH (which the authors can provide more evidence for – see point 4).

On the other hand, there are also obvious weaknesses:

• the novelty of pH-dependency of cell wall enzymes is questionable; several papers have already provided single-case examples;

• the claim that redundancy of cell wall enzymes boils down only to environment-specialized function is not something the current data can support. Specialization may be spatiotemporal (e.g. cell cycle), allow links to distinct protein machineries, allow additional functions… Redundancy could also ensure robustness of an essential process, targeted by many natural products in nature (antibiotics, T6SS effectors);

• the data for PBP1a taking control in high pH are weak/not convincing (see point 2), which undermines the main statement about clear pH division of bifunctional synthases;

• the statistics, the experimental description and sometimes the experiment setup do not warrant for the claims made (see points 1, 2 and 5);

• there are alternative ways to interpret the antibiotic resistance at low pH, which are independent of the PBP1b increased activity (see point 3).

Major points:

1) Credit should be given to authors for their effort to systematically detect growth defects of mutants in cell wall enzymes across different pH's (Figure 1). However, since most of the effects are small, statistics and experimental setups need be stronger/more transparent to substantiate claims. n=3 (Figure 1 and Figure 1—figure supplement 1) is very low to call 5% effects and to do t-tests (which assume normal distributions). It is also unclear how many measurements and what window of OD's are used for the exponential growth fit, and how good fits are at the end. Representative examples as a supp. figure would be extremely useful, in addition to providing the raw data and fits as supp. data. Last, there is no mention if authors look for/normalize plate effects (positional biases) when measuring for growth. This is very common in microtiter assays, and could very well be driving some of the small differences.

Since there are now a number of studies published in which the fitness of *E. coli* genome-wide mutant libraries has been probed across many different conditions (PMID 21185072, 27355376, 29769716), the authors may want to compare their results to these studies. pH is certainly one of these conditions. This way they can also look for further evidence of more conditional-specific roles for cell wall enzymes.

2) The are several pieces of data that make me doubt about the validity of the claims about the PBP1a importance/specialization in alkaline pH. I would suggest to remove (the emphasis from) this part, unless authors provide much stronger evidence.

In growth assays, the specific defect of the mrcA mutant in alkaline pH is questionable. Effect is marginal at ph8.2 in Figure 1A (mrcA seems to also have a very mild defect in all pHs in same figure) and Figure 2A mentions no n (done once?) and has no error bars to assess the significance of differences. Besides the two figures disagree; in 2A, the biggest difference is for ph~7 (difference gone for ph>8) and in 1A is at pH=8.2. Also speaking against of specialized role of PBP1A-LpoA at alkaline conditions, the lpoA mutant has the same small growth defect in all pH's (Figure 2C – if anything smaller at alkaline pHs)!

In microscopy, the cell death of mrcA is only transient (Figure 3B), so this cannot be the reason for the change in steady state growth as authors suggest (subsection “aPBP activity promotes cell wall integrity across pH environments”, second paragraph). Such an effect would result into also a transient growth defect in batch culture, which will be invisible the way authors do their experiments (as they back dilute 10^3 cells/ml so first doublings are in ODs that are below level of detection). Only way to explain batch growth rate defect (if it holds) is with an accompanying change in single cell growth rate, which authors can measure.

The data on PBP1A activity (Figure 5E) at different pH's are again not convincing. Signal is very low (btw this is completely the opposite from Figure 5D, where PBP1a has the strongest signal!), kinetics are fast (so many points and replicates are needed) and this seems to be the only replicate for measuring rates. It is also unclear how fits are done for Kobs (what type of nonlinear regression?), but they are definitely bad for ph5 (it seems as if a linear fit is shown). Besides more replicates with more points in the Bocillin assay, an in-vitro PBP1a assay may allow to see better kinetics and would strengthen any argument. Please keep in mind that pH7 that experiments are done for Figure 5E are not alkaline but neutral.

Overall, I find little evidence that PBP1a-LpoA takes over in alkaline conditions (as implied in text and Figure 7). Even if all effects are validated, at best it means that PBP1a/LpoA have a role under these conditions, but PBP1b/LpoB can compensate to a large degree.

3) Increased resistance to a sub-class of β-lactams at low pH is due to increased PBP1B activity. Although this could very well be (effect gone in mrcB mutant and controls in Figure 6—figure supplement 3C and D provide strong evidence), everything could be also explained with some of these drugs having decreased entry at low pH. Note that β-lactams have differential uptake and efflux preferences – Aztreonam for example needs OmpF to enter cell (see PMID – 29980614). Less drug in the cell could explain also why cells treated with Cephalexin or Mecillinam need more drug to change their morphology at pH5, although presumably the targets of the drugs, PBP3 and PBP2 do not change levels.

Measuring drug intracellular concentrations would be required to exclude such a scenario. Or alternatively some of the experiments in point 4 could strengthen the PBP1B increased activity in complexes. Also including more non-specific/broad β-lactams would (Carbenicillin, Amoxicillin) would help towards this direction. Note that Imipinem is not a selective PBP2 inhibitor- it targets many PBPs in addition to LD-transpeptidases.

4) The authors propose 3 models to rationalize how PBP1b increased activity at low pH could make cells more resistance to PBP2/3 inhibitors. They favor the one which PBP1b replaces PBP2 and PBP3 in the elongasome and divisome (as increased repair could for example not explain the need for increased Cephalexin to see filamentation at low pH). This scenario has some straightforward ramifications that are rather easy to test: PBP2/3 should be less active at low pH (Bocillin assay), PBP1b/2/3 dynamics may differ at low pH, cells should tolerate more depletion of PBP2/PBP3 at low pH. Also it could be interesting to test if *Salmonella* shows the same higher resistance at low pH (in PBP1b-dependent manner) when PBP3Sal is knocked out but not when there (because PBP3Sal is still as sensitive to PBP3 inhibitors).

5) Quantification and statistics suffer throughout the paper. Replicates are not always there (Figure 2A, Figure 5E) and many times too little for the statistics authors want to do: standard deviation from 2 replicates (5C, 6D), t-tests from sets of 3 experiments (Figure 1). Microscopy is not always quantified (Figure 5A, 6B/C), and the most worrisome is that panels do not always agree (Figure 1-2, or Figure 5D and 5E). All this should be amended for the conclusions to be on solid ground.

*Minor Comments:*

1) Introduction, second paragraph: the reason for the high sensitivity of the periplasm is not the different permeability of the two membranes; please rephrase.

2) Introduction, second paragraph: careful, there is a homeostatic control system: HdeA and HdeB are periplasmic chaperones induced and required during acidic conditions; quite some things known about their function. Also there is some literature on the requirement of other housekeeping periplasmic quality control enzymes during acid shock.

3) Introduction, fifth paragraph: why curiously?

4) Introduction, last paragraph: one "in" too much.

5) Throughout the text: "insertional deletions". Mutations come either from insertions or deletions. Since these are mutants from the KEIO collection, they are deletions.

6) Subsection “Identification of pH specialist cell wall synthases and hydrolases”, first paragraph: commensal *E. coli* (MG1655 is not even that) are not specialized for growing in urine. UPEC strains carry ~2,000 genes more than MG1655, so this leaves quite some room for other alkaline-specialized cell wall enzymes.

7) Subsection “aPBP activity ensures fitness across a wide pH range”, first paragraph: effect is not dramatic for mrcA.

8) Subsection “aPBP activity ensures fitness across a wide pH range”, second paragraph: how is p-val calculated with no replicates here?

9) Subsection “aPBP activity ensures fitness across a wide pH range”, second paragraph (Figure 2—figure supplement 1); could be informative to test pbpC double mutants with mrcA and mrcB to check if the effect of PBP1c is masked by any of the other two.

10) Subsection “aPBP substrate binding is pH-dependent”, first paragraph/Figure 5A: please quantify.

11) Subsection “pH-dependent PBP1b activity alters intrinsic resistance to PBP2 and PBP3 specific β-lactam antibiotics”, first paragraph: mrcB and lpoB mutants are very sensitive also to Cefsulodin, and a number of broad-acting cell wall enzymes. In contrast mecillinam effects are mild. These facts are hard to reconcile with line of thought that follows (see the second paragraph of the aforementioned subsection).

12) Subsection “pH-dependent PBP1b activity alters intrinsic resistance to PBP2 and PBP3 specific β-lactam antibiotics”, third paragraph/Figure 6D: other drugs and n>2 would be helpful to show that effect is specific.

13) Subsection “pH-dependent PBP1b activity alters intrinsic resistance to PBP2 and PBP3 specific β-lactam antibiotics”, last paragraph: referring to Figure 6E not 6D.

14) Subsection “Specialization role for aPBPs in cell wall integrity across environmental conditions”, first paragraph: if effects are direct on enzyme activity cannot be deduced from the Bocillin assay; many other upstream effects could compromise enzyme activity.

15) Subsection “Specialization role for aPBPs in cell wall integrity across environmental conditions”, second paragraph: first time I hear about depressed enzymes…

16) Subsection “Plasticity in cell wall metabolism potentiates intrinsic resistance to cell wall active antibiotics”, first paragraph: I find this highly speculative. Slow growth may as well be playing a more active role in resistance development. Also these microbes can become equally well resistant to broad cell wall inhibitors as they do to PBP3 inhibitors.

17) Subsection “Bacterial strains, plasmids, and growth conditions”: are strains re-transduced (once)? One clone used? Mutation checked?

18) Figure 3B: would be nice to see a longer experiment (past 2hrs); also can make points/schemes smaller to see error bars.

19) Figure 5D/E: inconsistent. PBP1a has the strong signal in d, quantified as low signal in e. PBP1b effect is ~ 2-fold at 15 min at Figure 5D, but looks less in 5E.

20) Figure 6A: What happens with Mecillinam at pH>7?

21) Figure 6—figure supplement 1: Azt is unstable at higher pH, but this does not correlate with increase in MIC (Figure 6)?!

22) Figure 6B/C: quantitative data would be more convincing.

23) Figure 6—figure supplement 3 legend: Define pH's you are comparing.

---

## [Author Response]

Reviewer #1:

This paper examines the pH-dependencies of PBP1a and PBP1b, arguing that each is "specialized" for a different pH. The data are relatively clean and soundly interpreted in most cases. Overall, the paper paints a compelling picture that the apparent redundancy of cell wall enzymes may reflect specialization for different environments. Below are several issues that I think the authors should respond to before publication – in some cases, this could involve significant new experiments, but I think such experiments would substantially add to the paper's mechanistic depth and long-term impact on the field.1) Figure 5C shows PBP1b levels – but what about PBP1a?

We thank the reviewer for this suggestion. We acquired a PBP1a antibody and have updated Figure 5C to include PBP1a levels across pH conditions. Normalizing to FtsZ levels as an internal loading control, this analysis revealed that both PBP1a and PBP1b levels are modestly reduced (~2 and 4-fold, respectively) during growth at pH 5.2 compared to pH 6.9. As noted in the text, this finding is consistent with the trend observed in Schmidt et al., 2016, by proteomic mass spectrometry when comparing cells grown at pH 6.0 to pH 7.0. In contrast, PBP1a and PBP1b levels were not significantly altered during growth at pH 8.2 compared to pH 6.9. This result strengthens our conclusion that changes in aPBP levels across pH environments are unlikely to contribute to the enzymes’ pH specificity.

2) Is there any significance to the clustering of GFP-PBP1a at low pH? I was sort of surprised that this observation wasn't followed up at all.

We agree with the reviewer this is an interesting observation that warrants further investigation in future studies. In the present revision, we have quantified PBP1a subcellular localization (see Figure 5—figure supplement 2), as suggested by reviewer 3. As expected, during growth at pH 7.0, GFP-PBP1b is enriched at the cell boundaries, similar to the localization profile previously reported for GFP-PBP1a (e.g. Paradis-Bleau, 2010). However, during growth at pH 5.0, GFP-PBP1a enrichment at the cell boundaries is reduced and instead forms irregular shaped puncta throughout the cell body. It is tempting to speculate that the reduction in peripheral GFP-PBP1a may reflect a reduction in the available/active PBP1a pool at the cell envelope and thus contribute to the cell’s reliance on PBP1b to provide the essential aPBP activity during growth in acidic media. This pH-dependent change in GFP-PBP1a localization may be a product of many factors, including but not limited to improper localization of a PBP1a activator (e.g. LpoA), sequestration of PBP1a by an unknown factor, an inactive conformational state of PBP1a, degradation of PBP1a or LpoA, or an inability of PBP1a to successfully traffic to the membrane. Rigorous assessment of each of these possibilities will be investigated in future studies.

3) PBP1b provides resistance to various antibiotics at low pH. But I don't really understand why. If the model is that PBP1a is more active and neutral and alkaline pH with PBP1b more active at low pH, why isn't PBP1a leading to comparable levels of resistance at the higher pHs? Do PBP1b and PBP1a, in fact, have different activities? I thought they were carrying out the same reaction, with one more active at low pH and the other more active at higher pHs, so the source of the difference in antibiotic susceptibility doesn't really make sense to me, but maybe I'm missing something here.

PBP1a and PBP1b share the same biochemical activities in vitro(i.e. transpeptidase and glycosyltransferase activity) and appear to be interchangeable for growth in vivo. At the same time, their activity is differentially required in vivo in response to different types of cell envelope stress, including antibiotic, osmotic, and mechanical challenge. In particular, PBP1b seems to be critical in response to these stressors, and PBP1a cannot fully compensate in the absence of PBP1b.

At present, it remains unclear why PBP1b is preferentially required for resistance to cell wall stress. Several explanations have been proposed. One possibility is that differences in enzymatic activity (e.g. differences in cross linking ability) and/or enzymatic efficiency of the class A PBPs contributes to their distinct roles in stress resistance. For example, Born et al., 2006, found that under optimal in vitroconditions PBP1b synthesized PG with 2x the amount of crosslinked peptides compared to PBP1a. While the relationship between the enzymatic properties of the class A PBPs and the PBP1b mutant antibiotic hypersensitivity has yet to be rigorously interrogated, there is little evidence to suggest enzymatic differences are playing a role in the pH-dependent resistance phenotype described here: comparison of *E. coli* PG composition at pH 7.5 and 5.0 did not reveal any differences in overall PG structure, including the percentage of crosslinked peptides (Peters et al., 2016). An alternative possibility is that the class A PBPs play unique roles in cell wall quality control. In support of this idea, in the time since our manuscript was submitted, work from Moré and colleagues (Moré et al., 2019) identified a role for LpoB/PBP1b (but not LpoA/PBP1a) in a so-called “PG repair machine” important for survival after outer membrane stress. It is possible that PBP1b may play a similar role in responding to β-lactam induced cell wall “damage”, and PBP1a cannot compensate. This model also remains to be tested.

Although of significant interest, the cause of the disparate contribution of PBP1a and PBP1b to β-lactam protection is not the focus of this investigation, so we have refrained on extensively commenting on it in the text. We have, however, added lines to the Discussion and throughout the text more explicitly distinguish between the class A PBPs’ overlapping roles in supporting cell growth (under standard culture conditions) and in stress/antibiotic resistance.

*4) I really think it's a lost opportunity to not provide more insight into the molecular basis for the pH differences in PBP1a and PBP1b. It appears that PBP1b activity can be reconstituted* in vitro*, including with LpoB and FtsN, so testing things* in vitro *at different pHs so testing things* in vitro *at different pHs seems potentially feasible. I would encourage the authors to at least consider whether something in this regard can be done to strengthen and enhance the paper.*

We thank the reviewer for encouraging us to investigate the impact of pH on the biochemical activity of PBP1a and PBP1b in the context of their known activators. To this end, we collaborated with Waldemar Vollmer’s group, which has extensive expertise in reconstituting activity of the class A PBPs in the presence of their cognate activators in vitro. The results of these experiments are included in Figure 5D, E, Figure 5—figure supplement 2, and Figure 5—figure supplement 3 and are discussed in the subsection “PBP1a localization and activity are impaired at low pH”. Overall, they support a model in which PBP1a activity is significantly reduced at pH 4.8, rendering the cell reliant on PBP1b activity in acidic media.

Briefly, we performed two PG synthesis assays. The first is an end point assay, in which purified enzymes and activators are reconstituted in micelles, supplied radiolabeled Lipid II precursor, and allowed to react for our hour. Synthesized PG is digested into muropeptides and resolved via HPLC. The fraction of unutilized Lipid II provides a qualitative metric for end-point glycosyltransferase activity, and the end-point transpeptidase activity can by quantified by the sum of the peaks corresponding to crosslinked products. This analysis revealed that PBP1b-LpoA has little activity at pH 4.8, while PBP1b retains similar activity across all pH conditions (Figure 5D, E).

This finding is further bolstered by the results of a continuous fluorescence assay (Figure 5—figure supplement 3). This assay measures glycosyltransferase reaction rate by quantifying the rate of polymerization of Dansyl-labled Lipid II substrate. Polymerization causes a decrease in fluorescence signal. As previously shown for PBP1b, both class A PBPs had a reduced polymerization rate in acidic media when assayed in the absence of their regulators. However, the presence of LpoB and FtsN significantly stimulated polymerization rate of PBP1b in all pH conditions (with the highest fold-change observed at pH 4.8); in contrast, LpoA failed to stimulate PBP1a under the same conditions. Note that the experimental conditions (e.g. temperature and enzyme concentration) vary between panels A and B due to technical limitations in accurately quantifying the rapid activity of PBP1b in the presence of its activators at 37 °C.

In attempt to distinguish whether the lack of PBP1a activity in acidic conditions is due to a decrease in affinity for LpoA, we performed a series of SRP experiments to measure the binding affinity of PBP1a and PBP1b to their cognate Lpo at pH 4.8, 6.9, and 8.2. Although affinity measurements could be made at pH 6.9 and 8.2, the Lpos both bound to the chip nonspecifically at pH 4.8 and impeded affinity calculations. Therefore, while it is evident that LpoB can stimulate PBP1b in acidic conditions (as evidenced by the GTase assay), it remains unclear whether the LpoA stimulation of PBP1a occurs at pH 4.8.

5) I wondered whether there are phylogenetic analyses to support the notion that PBP1a and PBP1b have evolved, at least in part, to allow cells to cope with different pH conditions. Are there strains of E. coli lacking PBP1b or even closely related species lacking it? If so, can they not tolerate low pHs and/or would their PBPs complement a PBP1a/1b double deletion only at certain pHs?

This is a very interesting hypothesis that we would love to test. However, previous phylogenetic analysis (Typas et al., 2010; see Figure 6) revealed that all γ-proteobacteria, including all examined *E. coli* and *Enterobacteriaceae* genomes, possess orthologs of PBP1a and PBP1b. While α and β proteobacteria only encode PBP1a orthologs, their evolutionary distance from *E. coli* MG1655 would complicate any conclusions that we would hope to make in doing such an analysis.

Minor Comments:Abstract, last sentence: Awkward sentence – rephrase for clarity.

Rewritten, as suggested.

"providing a force" maybe not "providing the force" in light of recent findings from KC Huang et al.

Modified in text in light of the recent finding by Rojas et al., 2018, that the outer membrane in Gram negative bacteria is also a load bearing element.

"in in" typo.

Modified in text.

Introduction, last sentence: Sort of a cryptic end to the Introduction – clarify why this is concerning and what the consequences are.

We agree. This sentence has been revised for clarity.

Subsection “Identification of pH specialist cell wall synthases and hydrolases”, first paragraph: The authors probably need to do an ANOVA test to control for multiple hypothesis testing. Even with a p < 0.01 to increase stringency, I think an ANOVA is the right choice here.

Agreed. We have re-analyzed our data using a one-way ANOVA with a p < 0.01 and normalized for multiple comparisons. With the exception of the Δ*mltB* mutant, the remaining five mutants’ growth rate defects remained significant by this analysis. Each the remaining mutants displays a consistent defect in mass doublings per hour across a discrete range of pH values (Figure 1—figure supplement 2), supporting our classification as ‘pH specialists’. In contrast, loss of MltB did not confer a consistent defect in DPH across pH conditions compared to the parental strain (see Author response image 1).

What does 'cytoskeletal-independent' mean? I don't think of either MreB or FtsZ as cytoskeletal elements anymore – reword for accuracy/precision.

We agree and have removed the adjective ‘cytoskeletal-independent’ to avoid confusion.

"optimized" is a strong word…maybe "tuned" would be better. I just always think it's tough to show something is truly optimal.

We agree and have replaced ‘optimized’ with ‘tuned’ as the reviewer suggested.

Subsection “aPBP activity promotes cell wall integrity across pH environment”, first paragraph: typo, "cells" not "cell".

Corrected.

Typo "previous a proteomic".

Corrected to ‘a previous proteomic’

Reviewer #2:

Most bacteria build a peptidoglycan cell wall that protects them from osmotic lysis. While enzymes required for the synthesis of the peptidoglycan precursor (lipid-linked disaccharide pentapeptide) are essential, many of the enzymes participating in the construction of the cell wall are not. In Escherichia coli, there is functional redundancy between two class a penicillin-binding proteins (PBPs) since removal of both PBP1a and PBP1b causes lethality. This has led to the idea that functional redundancy might be widespread among enzymes that synthesize and modify the cell wall, since removal of individual factors does not typically confer phenotypes. Here, Mueller and Levin challenge the view that these enzymes are simply redundant. They showed that the so-called "redundant" PBP1a and PBP1b enzymes have each specialized to function in different pH ranges. While PBP1a is dispensable in acidic conditions, PBP1b is essential; in contrast, in alkaline environments, PBP1b is dispensable, but PBP1a is required for optimal survival. This is a significant discovery and it is possible that, as the authors propose, similar specialization might apply to other enzymes involved in peptidoglycan synthesis and remodeling. Furthermore, the authors demonstrate that the intrinsic resistance to antibiotics can be different based on environmental pH.The manuscript is well written, experiments were well done, conclusions are justified, and the work reports significant findings.There is only one issue that should be easily addressed to strengthen one of the claims:The authors used a propidium iodide assay to quantify lysis. This assay measures the uptake of the dye not necessary lysis. Although the images of cells in Figure 3 and Figure 4 show that some cells have lysed, the authors should track loss of cytoplasmic contents (e.g. leakage of cytoplasmic GFP or LacZ) to demonstrate and quantify lysis. It would also be nice to complement lysis data with a kill curve (determine number of cells alive by CFUs after exposure to the specific pH). The authors could still leave the propidium iodide data to reflect envelope problems (but not necessarily lysis or even death).

We thank the reviewer for this suggestion. To complement this analysis, we transformed PBP1a and PBP1b defective cells with a plasmid that expresses *gfp* under an IPTG-inducible promoter and measured concurrent π staining and loss of cytoplasmic GFP signal by time lapse microscopy when cells were exposed to their respective ‘non-permissive’ pH conditions. All cells that stained PI+ (n = 419 for Δ*mrcB* and n = 789 for Δ*mrcA*) also lost cytoplasmic GFP signal within 1-2 frames (acquisitions were taken every 3 minutes). Population-level π staining/cytoplasmic GFP loss kinetics are now summarized in Figure 3—figure supplement 1. We do note that the lysis kinetics of the Δ*mrcB* mutant harboring the plasmid are slower compared to the untransformed strain (compare to Figure 3C), and qualitatively there appears to be a reduced rate of lysis via bulging. We speculate that this difference in phenotype may reflect a change in turgor pressure upon excess GFP production, but we did not investigate it further. Altogether, in combination with our new finding that single cell elongation rate is invariant among the mutants across pH conditions (a suggestion of reviewer 3), this new data strongly supports our model that the bulk culture growth rate defect of the mutants at their respective non-permissive pH condition is due to cell lysis.

Minor Comments:1) Delete repeated "in".

Modified in text.

2) Subsection “Identification of pH specialist cell wall synthases and hydrolases”, second paragraph: I think it should state "six" mutants. Not "seven".

The reviewer is correct. We have modified the text to five to reflect the new number of ‘hits’ after altering our statistical analysis and conducting further validation studies (see comments to reviewers 1 and 3).

3) Subsection “aPBP activity ensures fitness across a wide pH range”, first paragraph: Denome et al., 1999, might not be the best reference to cite the synthetic lethality of PBP1a and 1b because the pbp1a mutant used in that study contained additional mutations, as described in Meberg et al., J Bacteriol. 183:6148-9; PMID: 11567017. It would be more appropriate to cite the first report of the synthetic lethality: Suzuki et al., 1978.

We thank the reviewer for catching this. We have replaced the reference with the one suggested in the text.

Reviewer #3:

[…] Major points1) Credit should be given to authors for their effort to systematically detect growth defects of mutants in cell wall enzymes across different pH's (Figure 1). However, since most of the effects are small, statistics and experimental setups need be stronger/more transparent to substantiate claims. n=3 (Figure 1 and Figure 1—figure supplement 1) is very low to call 5% effects and to do t-tests (which assume normal distributions). It is also unclear how many measurements and what window of OD's are used for the exponential growth fit, and how good fits are at the end. Representative examples as a supp. figure would be extremely useful, in addition to providing the raw data and fits as supp. data. Last, there is no mention if authors look for/normalize plate effects (positional biases) when measuring for growth. This is very common in microtiter assays, and could very well be driving some of the small differences.

We thank the reviewer for encouraging us to provide more information on our experimental set up and data analysis, and we wholeheartedly agree that growth rate measurements can be highly sensitive to uncontrolled variables. We have rigorously designed our screen to minimize sources of uncontrolled variation. We have added additional information in the text to clarify our experimental and analysis pipeline, including subsection “Identification of pH specialist cell wall synthases and hydrolases”, in the Materials and methods, Figure 1—figure supplement 1, Figure 1—figure supplement 2, and Supplementary file 3 to address these concerns.

As we state in the original version of the manuscript, growth rates were determined by least-squares fitting of growth curves between the OD600 values of 0.005-0.1 for three independent replicates per mutant. This analysis was performed in R. For transparency, we now provide sample growth curves, fits, and fit statistics for a subset of our mutants in Figure 1—figure supplement 1. Best fit lines with R^2^ values of < 0.95 were excluded from further analysis. In addition to the representative curves and fits, we have included the growth rates (as measured in mass doublings per hour +/- standard deviation) for all tested mutants at pH 4.8, 6.9, and 8.2 in Supplementary file 3 (n=3 for all measurements). No positional effects were observed. As suggested by reviewer 1, we re-assessed significance of our hits using a one-way ANOVA, normalized for multiple comparisons; 5 of the 6 original hits remained significantly attenuated with this analysis.

To address the concern that some of our hits confer relatively small reductions in growth rate, we added an additional layer of validation to our analysis. We reasoned that a consistent growth defect across a discrete range of pH values would be a rigorous way to validate bona fide hits as opposed to statistical aberrations (analogous to what we had previously done for PBP1a and PBP1b-defective cells). To this end, we compared the growth rate of each mutant to the WT across a range of pH values (pH 4.8-8.4). Six biological replicates spread across at least two days were performed. Excitingly, all five of the mutants tested demonstrated consistent, statistically significant growth rate defects across a contiguous range of pH values (see Figure 1—figure supplement 2 and Figure 2). This data is now plotted in both absolute doublings per hour as well as% doublings per hour of the parental strain to allow for easier comparison between the mutants’ growth across different pH conditions (see point #2). This analysis strongly suggests each of the mutants pulled out in our screen exhibits a specific and reproducible pH sensitivity. We would like to further note that relatively small differences in mutant growth rate may lead to significant fitness defects in natural environments with competition for limited resources.

Since there are now a number of studies published in which the fitness of E. coli genome-wide mutant libraries has been probed across many different conditions (PMID 21185072, 27355376, 29769716), the authors may want to compare their results to these studies. pH is certainly one of these conditions. This way they can also look for further evidence of more conditional-specific roles for cell wall enzymes.

We thank the reviewer for making us aware of these resources. Excitingly, loss of function mutants in the genes encoding PBP1b and MepS both have acid-specific growth defects for colony formation (PMID 21185072). We have added a sentence to the text (subsection “Class A PBP activity ensures fitness across a wide pH range”) to reflect this.

2) The are several pieces of data that make me doubt about the validity of the claims about the PBP1a importance/specialization in alkaline pH. I would suggest to remove (the emphasis from) this part, unless authors provide much stronger evidence.In growth assays, the specific defect of the mrcA mutant in alkaline pH is questionable. Effect is marginal at ph8.2 in Figure 1A (mrcA seems to also have a very mild defect in all pHs in same Fig) and Figure 2A mentions no n (done once?) and has no error bars to assess the significance of differences. Besides the two figures disagree; in 2A, the biggest difference is for ph~7 (difference gone for ph>8) and in 1A is at pH=8.2. Also speaking against of specialized role of PBP1A-LpoA at alkaline conditions, the lpoA mutant has the same small growth defect in all pH's (Figure 2C- if anything smaller at alkaline pHs)!In microscopy, the cell death of mrcA is only transient (Figure 3B), so this cannot be the reason for the change in steady state growth as authors suggest (subsection “aPBP activity promotes cell wall integrity across pH environments”, second paragraph). Such an effect would result into also a transient growth defect in batch culture, which will be invisible the way authors do their experiments (as they back dilute 10^3 cells/ml so first doublings are in ODs that are below level of detection). Only way to explain batch growth rate defect (if it holds) is with an accompanying change in single cell growth rate, which authors can measure.The data on PBP1A activity (Figure 5E) at different pH's are again not convincing. Signal is very low (btw this is completely the opposite from Figure 5D, where PBP1a has the strongest signal!), kinetics are fast (so many points and replicates are needed) and this seems to be the only replicate for measuring rates. It is also unclear how fits are done for Kobs (what type of nonlinear regression?), but they are definitely bad for ph5 (it seems as if a linear fit is shown). Besides more replicates with more points in the Bocillin assay, an in-vitro PBP1a assay may allow to see better kinetics and would strengthen any argument. Please keep in mind that pH7 that experiments are done for Figure 5E are not alkaline but neutral…Overall, I find little evidence that PBP1a-LpoA takes over in alkaline conditions (as implied in text and Figure 7). Even if all effects are validated, at best it means that PBP1a/LpoA have a role under these conditions, but PBP1b/LpoB can compensate to a large degree.

While we acknowledge that the growth defect for the PBP1a-defective strain in neutral/alkaline media is modest in comparison to the complete loss of growth of the PBP1b-defective strain at pH 4.8, we stand by our assertion that PBP1a is required for maximal fitness in this pH range.

At least part of this reviewer’s concern seems to stem from misinterpretation of the data extrapolated from the figures. For example, although the reviewer claims there is a discrepancy between Figures 1A and 2A in terms of which pH PBP1a-deficient cells have the greatest maximal growth defect, in both of these figures, PBP1a-deficient cells have a greater growth defect at pH 8.2 than 6.9. To avoid confusion, we have added a table denoting the mean growth rate +/- SD for each mutant tested in our original screen (Supplementary file 3), as well as panels in Figure 2 and Figure 1—figure supplement 2 depicting the% WT growth (DPH for mutant/DPH of wild-type x 100). We hope that this representation of the data allows for more accurate comparisons among the mutants and pH conditions. We have also updated our figure legends to reflect the n value for each experiment. Figure 2A-C, for example, has an n = 6 for each strain (3 replicates were presented in the original draft of the manuscript, but the n was increased to 6 in response to this reviewer’s concern in point #1). Our findings in these experiments reveal a reproducible and significant defect for PBP1a from pH 6.9-8.2.

In this revision, we also repeated the pH sensitivity testing of the *lpo* mutants at an extended pH range. Cells defective for LpoA display a comparable range and magnitude of growth defect compared to PBP1a defective cells. (updated Figure 2C; Figure 2—figure supplement 2). These data provide additional support for a specific role for PBP1a in these conditions.

In response to the reviewer’s concern that the transient lytic phenotype for the mutant defective for PBP1a could not underlie the bulk culture growth defect, we examined lysis kinetics in liquid culture to more closely mimic the conditions of our initial screen. Briefly, cells were cultured to mid-exponential phase in pH 6.9 media, back-diluted to an OD600 of 0.005 at pH 8.2 and sampled at various time points for propidium iodine (PI) staining via fluorescence microscopy. As shown in Figure 3—figure supplement 1D, up to 10% of the cells lysed up to 3 hours post-alkaline shock with no observable recovery. Although we hoped to extend the experiment beyond 3 hours, at cell densities with OD600 values > 1, the buffering capacity of the media was overwhelmed, and the media began to decrease in pH. In light of this finding, we speculate that the recovery phenotype observed on the agarose pad may reflect a decrease in local pH upon increasing cell density, protecting the cells against further lysis. In tandem, we measured single cell elongation rate for each of the mutants and parental strain from our time lapse videos at pH 4.5 and 8.0 in SuperSegger, a MATLAB based cell segmentation program. As shown in Figure 3A-B, no differences in elongation rate were observed between the strains at each pH, implicating lysis as the sole source of the bulk culture growth defect.

The reviewer has valid concerns about the Bocillin assays; our signal is low and inconsistent (likely in part due to a technical limitation in using an LED-based imager instead of a fluorescence scanner with higher sensitivity, as is typical for these experiments), and these assays fail to differentiate between intrinsic and extrinsic effects on enzyme activity. In consideration of this, we have removed them from the manuscript and instead replaced them with in vitroassays of biochemical activity conducted in collaboration with Waldemar Vollmer’s group, as suggested by reviewer 1. Briefly, we find that PBP1a has negligible glycosyltransferase and transpeptidase activity at pH 4.8, which causes the cell to be reliant on PBP1b activity in this pH range (Figure 5D, E; Figure 5—figure supplement 3; Figure 5—figure supplement 4). Although we do not observe any differences in PBP1b activity that account for the preference for PBP1a in more alkaline conditions, these phenotypes may be too subtle to capture in these assays, or alternatively, differences in the aPBP’s subcellular localization and/or interactions partners may also contribute to their pH specialization. Future work will be necessary to evaluate these possibilities.

3) Increased resistance to a sub-class of β-lactams at low pH is due to increased PBP1B activity. Although this could very well be (effect gone in mrcB mutant and controls in Figure 6—figure supplement 3C and d provide strong evidence), everything could be also explained with some of these drugs having decreased entry at low pH. Note that β-lactams have differential uptake and efflux preferences – Aztreonam for example needs OmpF to enter cell (see PMID – 29980614). Less drug in the cell could explain also why cells treated with Cephalexin or Mecillinam need more drug to change their morphology at pH5, although presumably the targets of the drugs, PBP3 and PBP2 do not change levels.Measuring drug intracellular concentrations would be required to exclude such a scenario. Or alternatively some of the experiments in point 4 could strengthen the PBP1B increased activity in complexes. Also including more non-specific/broad β-lactams would (Carbenicillin, Amoxicillin) would help towards this direction. Note that Imipinem is not a selective PBP2 inhibitor- it targets many PBPs in addition to LD-transpeptidases.

These β-lactams act on their targets in periplasmic space, not in the cytoplasm. We assume the reviewer meant to ask us to measure periplasmic concentration of the compounds. To our knowledge, there is no direct method to measure this, and to design one would require significant innovation beyond the scope of the current study.

We have added additional data to the revision to support our conclusion that changes in outer membrane permeability are unlikely to underlie the low pH-dependent resistance phenotype.

A) Additional compounds we have tested and included in this revision match our previously observed results. At the suggestion of the reviewer, we added an additional broad-spectrum antibiotic (amoxicillin) and two additional PBP2 inhibitors (meropenem and doripenem). As expected, MG1655 exhibited reduced susceptibility to the PBP2 inhibitors at pH values < 6.0. In contrast, the MIC to amoxicillin was only mildly elevated (2-fold) at pH values < 5.0. (compared to >8-fold for the majority of PBP2/PBP3-specific compounds). Importantly, while meropenem and doripenem do target the L,D-transpeptidases (similar to imipenem, as the reviewer noted), cells that lack three or six L,D-transpeptidases still exhibited a comparable increase in the drugs at pH 5.0 (Figure 6—figure supplement 4E), indicating that these enzymes do not play a role in this phenotype. As an aside, we have removed imipenem from the study because we found it to be unstable at pH 4.5.

B) As a separate means of addressing the contribution of OM permeability to our phenotype, we treated cells with sub-inhibitory concentrations of polymyxin B, a compound which forms pores in the outer membrane. We reasoned that this treatment would non-specifically compromise outer membrane integrity in cells independent of pH. Thus, if our pH-dependent changes in MIC were due to differences in outer membrane integrity, there would be no difference in CEX MIC at pH 7 and pH 5.0. Instead, we observed the same fold-change in MIC as cells not treated with polymyxin B. Importantly, the concentrations of Polymixin B used in these assays decrease the cells’ CEX MIC at both pH values, indicating comparable outer membrane disruption (Figure 6—figure supplement 3C). This data offers strong evidence that low pH dependent resistance cannot be explained solely by changes in outer membrane permeability.

4) The authors propose 3 models to rationalize how PBP1b increased activity at low pH could make cells more resistance to PBP2/3 inhibitors. They favor the one which PBP1b replaces PBP2 and PBP3 in the elongasome and divisome (as increased repair could for example not explain the need for increased Cephalexin to see filamentation at low pH). This scenario has some straightforward ramifications that are rather easy to test: PBP2/3 should be less active at low pH (Bocillin assay), PBP1b/2/3 dynamics may differ at low pH, cells should tolerate more depletion of PBP2/PBP3 at low pH. Also it could be interesting to test if Salmonella shows the same higher resistance at low pH (in PBP1b-dependent manner) when PBP3Sal is knocked out but not when there (because PBP3Sal is still as sensitive to PBP3 inhibitors).

We agree with the reviewer that this is a very interesting question and appreciate their suggestions on ways in which we can test this model in future studies. However, we believe rigorously addressing this model necessitates a thorough investigation into the effect of pH on the composition and the activity of both the elongation and division machinery, which is beyond the scope of the current manuscript. Consequently, we have significantly revised the Discussion to decrease the emphasis on this point.

5) Quantification and statistics suffer throughout the paper. Replicates are not always there (Figure 2A, Figure 5E) and many times too little for the statistics authors want to do: standard deviation from 2 replicates (5C, 6D), t-tests from sets of 3 experiments (Figure 1). Microscopy is not always quantified (Figure 5A, 6B/C), and the most worrisome is that panels do not always agree (Figure 1-2, or Figure 5D and 5E). All this should be amended for the conclusions to be on solid ground.

As the reviewer suggested, we have re-evaluated our statistical test choice (Figure 1; Figure 2D), added additional replicates (Figure 2A-C; Figure 5C; Figure 6D), and quantified our microscopy (Figure 5A, B; Figure 6B, C).

Minor Comments:1) Introduction, second paragraph: the reason for the high sensitivity of the periplasm is not the different permeability of the two membranes; please rephrase.

It is a widely held belief that differential membrane permeability contributes to pH and osmotic sensitivity of the periplasm. The outer membrane contains porins, including OmpC and OmpF, that are permissive to ions, protons and water; these porins are not present in the inner membrane, and thus the inner membrane is often considered the major permeability barrier for Gram negative bacteria. If the reviewer can point us to references that refute this point or elaborate on why they took issue with this statement, we are happy to revise accordingly.

2) Introduction, second paragraph: careful, there is a homeostatic control system: HdeA and HdeB are periplasmic chaperones induced and required during acidic conditions; quite some things known about their function. Also there is some literature on the requirement of other housekeeping periplasmic quality control enzymes during acid shock.

We have removed the clause “… in the absence of a homeostatic control system.”

3) Introduction, fifth paragraph: why curiously?

Rewritten.

4) Introduction, last paragraph: one "in" too much

Amended.

5) Throughout the text: "insertional deletions". Mutations come either from insertions or deletions. Since these are mutants from the KEIO collection, they are deletions.

We have changed “insertional deletions” to deletions in all instances in the text.

6) Subsection “Identification of pH specialist cell wall synthases and hydrolases”, first paragraph: commensal E. coli (MG1655 is not even that) are not specialized for growing in urine. UPEC strains carry ~2,000 genes more than MG1655, so this leaves quite some room for other alkaline-specialized cell wall enzymes.

We agree with the reviewer that the pangenome of UPEC and intestinal *E. coli* exceeds that of MG1655, and importantly, we never claim to identify an exhaustive list of pH-specialized cell wall enzymes. While it would be interesting to investigate pH-specialized cell wall enzymes in UPEC, selecting a representative strain to conduct this analysis would be challenging for several reasons. In particular, UPEC strains exhibit remarkable genetic diversity and lack a conversed genetic signature for urovirulence (Schreiber IV et al., 2017).

7) Subsection “aPBP activity ensures fitness across a wide pH range”, first paragraph: effect is not dramatic for mrcA.

The word ‘dramatic’ has been removed.

8) Subsection “aPBP activity ensures fitness across a wide pH range”, second paragraph: how is p-val calculated with no replicates here?

As addressed above, n = 3 for this experiment.

9) Subsection “aPBP activity ensures fitness across a wide pH range”, second paragraph (Figure 2—figure supplement 1); could be informative to test pbpC double mutants with mrcA and mrcB to check if the effect of PBP1c is masked by any of the other two.

As suggested, we constructed these mutants and tested for pH-dependent growth rate defects from pH 4.8-8.4. Loss of PBP1c did not exacerbate any of the growth defects, indicating its effects are not masked by the presence of the other aPBPs (Figure 2—figure supplement 1).

10) Subsection “aPBP substrate binding is pH-dependent”, first paragraph/Figure 5A: please quantify.

As suggested, we have now quantified PBP1a localization (see Figure 5—figure supplement 2). See comments to reviewer 1 for additional commentary on localization profile.

11) Subsection “pH-dependent PBP1b activity alters intrinsic resistance to PBP2 and PBP3 specific β-lactam antibiotics”, first paragraph: mrcB and lpoB mutants are very sensitive also to Cefsulodin, and a number of broad-acting cell wall enzymes. In contrast mecillinam effects are mild. These facts are hard to reconcile with line of thought that follows (see the second paragraph of the aforementioned subsection).

This section has been removed from the text.

12) Subsection “pH-dependent PBP1b activity alters intrinsic resistance to PBP2 and PBP3 specific β-lactam antibiotics”, third paragraph/Figure 6D: other drugs and n>2 would be helpful to show that effect is specific.

We repeated this experiment with fresh urine (n=3) and results are shown in an updated version of Figure 6D.

13) Subsection “pH-dependent PBP1b activity alters intrinsic resistance to PBP2 and PBP3 specific β-lactam antibiotics”, last paragraph: referring to Figure 6E not 6D.

Corrected.

14) Subsection “Specialization role for aPBPs in cell wall integrity across environmental conditions”, first paragraph: if effects are direct on enzyme activity cannot be deduced from the Bocillin assay; many other upstream effects could compromise enzyme activity.

We agree. As previously mentioned, we have removed the Boc-FL binding assays from the manuscript.

15) Subsection “Specialization role for aPBPs in cell wall integrity across environmental conditions”, second paragraph: first time I hear about depressed enzymes…

Revised.

16) Subsection “Plasticity in cell wall metabolism potentiates intrinsic resistance to cell wall active antibiotics”, first paragraph: I find this highly speculative. Slow growth may as well be playing a more active role in resistance development. Also these microbes can become equally well resistant to broad cell wall inhibitors as they do to PBP3 inhibitors

We have revised this section to soften the language.

17) Subsection “Bacterial strains, plasmids, and growth conditions”: are strains re-transduced (once)? One clone used? Mutation checked?

Two transductants/clones were tested per strain. Positive hits were confirmed by diagnostic PCR. The Materials and methods section has been updated to reflect this.

18) Figure 3B: would be nice to see a longer experiment (past 2hrs); also can make points/schemes smaller to see error bars.

Unfortunately, after 2 hours cells grown at pH 8.0 become too dense on the agarose pads to accurately quantify. As previously mentioned, our liquid culture experiment with the PBP1a-defective strain extends to 3 hours.

19) Figure 5D/E: inconsistent. PBP1a has the strong signal in d, quantified as low signal in e. PBP1b effect is ~ 2-fold at 15 min at Figure 5D, but looks less in 5E.

As previously mentioned, we have removed the Boc-FL experiments that this comment refers to.

20) Figure 6A: What happens with Mecillinam at pH>7?

At present, is unclear how alkaline pH is conferring mecillinam resistance. We chose not to investigate this further as it was not conserved across other PBP2 inhibitors. However, the mechanism of high pH-dependent resistance is likely distinct from that observed in acidic conditions, as rod shape is not similarly preserved (see Figure 6—figure supplement 2).

21) Figure 6—figure supplement 1: Azt is unstable at higher pH, but this does not correlate with increase in MIC (Figure 6)?!

We anticipate that this “discrepancy” is due to the fast-acting lytic activity (<2 hours) of most β-lactams, including PBP3 inhibitors (see Fredborg et al. BCM Microbiol., 2015 for example lysis kinetics for piperacillin). Therefore, AZT treatment likely kills cells faster than it is hydrolyzed and inactivated at pH 8.0.

22) Figure 6B/C: quantitative data would be more convincing.

As suggested, we have now quantified cell length and aspect ratio for cells treated with sub-MIC levels of cephalexin and mecillinam, respectively (see revised Figure 6B/C).

23) Figure 6—figure supplement 3 legend: Define pH's you are comparing.

Amended.